

# Non-stomatal exchange in ammonia dry deposition models: Comparison of two state-of-the-art approaches

Frederik Schrader[1], Christian Brümmer[1], Chris R. Flechard[2], Roy J. Wichink Kruit[3], Margreet C. van Zanten[3], Undine Richter[1], Arjan Hensen[4], Jan Willem Erisman[5,6]

[1]Thünen Institute of Climate-Smart Agriculture, Braunschweig, Germany
[2]Instiute National de la Recherche Agronomique (INRA), Agrocampus Ouest, UMR1069 SAS, Rennes, France
[3]National Institute for Public Health and the Environment (RIVM), Bilthoven, The Netherlands
[4]Energy research Centre of the Netherlands (ECN), Petten, The Netherlands
[5]Cluster Earth and Climate, Department of Earth Sciences, Vrije Universiteit Amsterdam, Amsterdam, The Netherlands
[6]Louis Bolk Institute, Driebergen, The Netherlands

*Correspondence to*: Frederik Schrader (frederik.schrader@thuenen.de)

**Abstract.** The accurate representation of bidirectional ammonia ($NH_3$) biosphere-atmosphere exchange is an important part of modern air quality models. However, the cuticular (or external leaf surface) pathway, as well as other non-stomatal ecosystem surfaces, still pose a major challenge of translating our knowledge into models. Dynamic mechanistic models including complex leaf surface chemistry have been able to accurately reproduce measured bidirectional fluxes in the past, but their computational expense and challenging implementation into existing air quality models call for steady-state simplifications. We here qualitatively compare two semi-empirical state-of-the-art parameterizations of a unidirectional non-stomatal resistance ($R_w$) model after Massad et al. (2010), and a quasi-bidirectional non-stomatal compensation point ($\chi_w$) model after Wichink Kruit et al. (2010), with $NH_3$ flux measurements from five European sites. In addition, we tested the feasibility of using backward-looking moving averages of air $NH_3$ concentrations as a proxy for prior $NH_3$ uptake and driver of an alternative parameterization of non-stomatal emission potentials ($\Gamma_w$) for bidirectional non-stomatal exchange models. Results indicate that the $R_w$-only model has a tendency to underestimate fluxes, while the $\chi_w$ model mainly overestimates fluxes, although systematic underestimations can occur under certain conditions, depending on temperature and ambient $NH_3$ concentrations at the site. The proposed $\Gamma_w$ parameterization appears to have potential for improvement, but cannot be recommended for use in large scale simulations in its present state due to large uncertainties. As an interim solution for improving flux predictions, we recommend to reduce the minimum allowed $R_w$ and the temperature response parameter in the unidirectional model and to revisit the temperature dependent $\Gamma_w$ parameterization of the bidirectional model.

## 1 Introduction

Reactive nitrogen ($N_r$) deposition can contribute to a number of adverse environmental impacts, including ecosystem acidification, shifts in biodiversity, or climate change (Erisman et al., 2013). Breakthroughs in the measurement of



biosphere-atmosphere exchange of ammonia ($NH_3$), the major constituent of $N_r$ (Sutton et al., 2013), have been made in the recent past with the rising availability of high-frequency measurement devices that can be used within the eddy covariance method (e.g. Famulari et al., 2004; Ferrara et al., 2012; Richter et al., 2016), and a large body of flux measurements using other measurement techniques, e.g. the aerodynamic gradient method, has emerged from large-scale projects such as

NitroEurope (Sutton et al., 2011). These measurements, however, are usually only representative for a specific location and difficult to interpolate in space. Surface-atmosphere exchange schemes that predict ammonia exchange fluxes from measured or modeled concentrations and micrometeorological conditions are used on both the local scale and within large-scale chemical transport models (CTMs). Following the discovery of the ammonia compensation point (Farquhar, 1980), these models are nowadays able to reproduce bidirectional exchange fluxes, i.e. both emission and deposition of ammonia, and

typically feature at least a stomatal and a non-stomatal leaf surface pathway. The addition of a soil- or leaf litter pathway by Nemitz et al. (2001) has been recognized as an optimal compromise between model complexity and accuracy of the flux estimates (Flechard et al., 2013), although some uncertainties in the treatment of the ground layer still prevail.

While the representation of the stomatal pathway has received much attention in the literature due to its importance not only for ammonia, but also for a large number of other atmospheric constituents, especially carbon dioxide ($CO_2$) and water vapor

($H_2O$) (e.g. Jarvis, 1976; Farquhar and Sharkey, 1982; Ball et al., 1987), modeling non-stomatal exchange is still subject to considerable uncertainty (Burkhardt et al., 2009). Ammonia is highly soluble in water and thus readily deposits to water layers on the leaf cuticle, and on any other environmental surface, following precipitation events, condensation of water vapor, or due to the presence of hygroscopic particles on the surface. This characteristic behavior is often modeled using relative humidity response functions as a proxy for canopy wetness (e.g. Sutton and Fowler, 1993; Erisman et al. 1994). A

self-limiting effect of ambient ammonia concentrations on the deposition process, due to saturation effects and an increase in surface pH, has been observed in experiments (Jones et al, 2007a,b; Cape et al., 2008) and implemented in some non-stomatal exchange models (e.g. Wichink Kruit et al., 2010). Additionally, re-emission events during evaporation of leaf surface water layers have been measured in the field, which hints at the limits of these classical static and unidirectional approaches (Wyers and Erisman, 1998). Sutton et al. (1998) and Flechard et al. (1999) have successfully reproduced

measurements of these events on the field scale by modeling the water films as charged capacitors for ammonia emissions; however, these models need complex dynamic leaf chemistry modules which drastically increase computational expense and necessary input variables and consequently limit their applicability in large scale simulations. Wichink Kruit et al. (2010) developed a static hybrid-model featuring a non-stomatal compensation point approach in order to simplify the model calculations and as an important step towards the use of a bidirectional non-stomatal exchange paradigm within large scale

CTMs. In this paper, we compare the performance of two state-of-the-art parameterizations of non-stomatal exchange: The unidirectional approach of Massad et al. (2010) and the quasi-bidirectional approach of Wichink Kruit et al. (2010). We highlight strengths and weaknesses of both approaches and apply them to five measurement sites in Germany, the UK, the Netherlands and Switzerland. Predicted (effective) non-stomatal resistances are compared to those inferred from night-time flux measurements, when stomata are mostly closed and the contribution of the non-stomatal pathway to the total observed





flux is dominant. In addition, we investigate the potential of parameterizing a bidirectional non-stomatal exchange model by testing backwards-looking moving averages of air ammonia concentrations as a proxy for prior ammonia inputs into the ecosystem, eliminating the need for dynamic or iterative flux-based approaches with the use of a readily available, easy-to-calculate and easy-to-implement metric.

## 5  2 Methods

### 2.1 Bidirectional ammonia exchange models

Ammonia dry deposition is typically modeled using an electrical analogy based on a network of serial and parallel resistances. The two-layer model structure introduced by Nemitz et al. (2001) has been recognized as a good compromise between model complexity, ease of use and accuracy of the resulting exchange fluxes (Flechard et al., 2013), and it is the

foundation for the parameterization of Massad et al. (2010) that is used throughout this study. However, in the Massad et al. (2010) formulation, the second (soil / leaf-litter) layer is essentially switched off for semi-natural ecosystems and managed ecosystems outside of management events, because soil emissions are expected to be negligible in these cases. We therefore focus on the one-layer big-leaf model (Fig. 1) in this paper.

In the simplest form, the canopy resistance model (e.g. Wesely, 1989; Erisman and Wyers, 1993), surface-atmosphere-fluxes

are limited by three resistances in series: The aerodynamic resistance $R_a\{z-d\}$ (s m$^{-1}$) at the reference height $z-d$ (m) (where $z$ (m) is the measurement height above ground and $d$ (m) is the zero-plane displacement height), the quasi-laminar boundary layer resistance $R_b$ (s m$^{-1}$), and the canopy resistance $R_c$ (s m$^{-1}$). While $R_a\{z-d\}$ and $R_b$ are mainly dependent on micrometeorological conditions, surface roughness and chemical properties of the compound of interest, $R_c$ is directly dependent on the characteristics of the vegetated surface. The inverse of the sum of these three resistances is called the

deposition velocity, $v_d\{z-d\}$ (m s$^{-1}$).

$R_c$ is further split into a stomatal pathway with the stomatal resistance $R_s$ (s m$^{-1}$), and a non-stomatal (or cuticular) pathway with the non-stomatal resistance $R_w$ (s m$^{-1}$) (e.g. Erisman et al., 1994; Sutton et al., 1998). Stomatal exchange is usually modeled bidirectionally for ammonia in field scale studies and some CTMs, i.e. it is assumed that there is a non-zero gaseous ammonia concentration $\chi_s$ (µg m$^{-3}$) in equilibrium with dissolved ammonia in the apoplastic fluid. This concentration is

often called the stomatal compensation point, although strictly speaking the compensation point is only met when $\chi_s$ is approximately equal to the air ammonia concentration at the reference height $\chi_a\{z-d\}$ (ug m$^{-3}$) and consequently the net flux $F_t$ (ng m$^{-2}$ s$^{-1}$) is zero (Farquhar, 1980). The non-stomatal pathway is modeled unidirectionally in many parameterizations, i.e. the gaseous ammonia concentration in equilibrium with the solution on the external leaf surfaces $\chi_w$ (µg m$^{-3}$) is assumed to be zero, although observational evidence indicates that this pathway is in fact bidirectional as well

(e.g. Neirynck and Ceulemans, 2008). A canopy compensation point, $\chi_c$ (µg m$^{-3}$), that integrates these two pathways can be calculated as (e.g. Sutton et al., 1995; modified to include $\chi_w$):



$$\chi_c = \frac{\chi_a\{z-d\}\cdot(R_a+R_b)^{-1}+\chi_s\cdot R_s^{-1}+\chi_w\cdot R_w^{-1}}{(R_a\{z-d\}+R_b)^{-1}+R_s^{-1}+R_w^{-1}}, \tag{1}$$

and the total net flux of ammonia to or from the ecosystem, $F_t$ (ng m$^{-2}$ s$^{-1}$) as

$$F_t = -\frac{\chi_a\{z-d\}-\chi_c}{R_a\{z-d\}+R_b}, \tag{2}$$

where by convention negative fluxes indicate deposition towards the surface and positive fluxes indicate emission. This is typically done on a half-hour basis for consistency with flux measurement practices. $R_a\{z-d\}$ and $R_b$ are here modeled after Garland (1977) as:

$$R_a\{z-d\} = \frac{u\{z-d\}}{u_*^2} - \frac{\Psi_H\{\frac{z-d}{L}\}-\Psi_M\{\frac{z-d}{L}\}}{k\cdot u_*}, \tag{3}$$

and

$$R_b = u_*^{-1}\left[1.45\cdot\left(\frac{z_0\cdot u_*}{v_{air}}\right)^{0.24}\cdot\left(\frac{v_{air}}{D_{NH_3}}\right)^{0.8}\right], \tag{4}$$

where $u\{z-d\}$ (m s$^{-1}$) is the wind speed at the reference height, $u_*$ (m s$^{-1}$) is the friction velocity, $L$ (m) is the Obukhov length, $k$ (–) is the von Kármán constant ($k = 0.41$), $\Psi_H$ (–) and $\Psi_M$ (–) are the integrated stability corrections for entrained scalars and momentum, respectively, after Webb (1970) and Paulson (1970), $z_0$ (m) is the roughness length, $v_{air}$ (m$^2$ s$^{-1}$) is the kinematic viscosity of air, and $D_{NH_3}$ (m$^2$ s$^{-1}$) is the molecular diffusivity of ammonia in air. $R_s$ can be modeled using at least a light and temperature response function (e.g. Weseley, 1989), often with additional reduction factors accounting for vapor pressure deficit, soil moisture and other environmental variables (e.g. Emberson et al., 2000). However, this study focuses on nighttime fluxes when non-stomatal fluxes are assumed to be dominant. If $R_s$ is assumed to approach infinity at during nighttime, all terms involving $R_s$ in Eq. (1) collapse to zero.

## 2.2 Most recent non-stomatal resistance parameterizations

*(i) Massad et al. (2010)*

Based on an extensive meta-analysis, Massad et al. (2010) derived a parameterization (henceforth referred to as *MNS*) for a unidirectional non-stomatal pathway model (i.e. $\chi_w = 0$) that models the effect of the pollution climate by incorporating a so-called acid ratio, $AR$ (–), to scale the minimum allowed $R_w$. It is defined as the molar ratio of average total acid/NH$_3$ concentrations, $AR = (2[SO_2] + [HNO_3] + [HCl])/[NH_3]$ and is an extension of the classical [SO$_2$]/[NH$_3$] co-deposition proxy concept following the decline of SO$_2$ emissions in Europe during the last few decades (e.g. Erisman et al., 2001). In addition, effects of leaf area index $LAI$ (m$^2$ m$^{-2}$) and temperature $T$ (°C) are modeled following Zhang et al. (2003) and Flechard et al. (2010), respectively. With all corrections $R_w$ is given as:

$$R_{w,MNS} = R_{w,min}\cdot AR^{-1}\cdot e^{a\cdot(100-RH)}\cdot\frac{e^{\beta\cdot|T|}}{\sqrt{LAI}}, \tag{5}$$





where $R_{\text{w,min}} = 31.5$ s m$^{-1}$ is the 'baseline' minimum $R_{\text{w}}$, $a$ (–) is an empirical ecosystem-specific parameter ranging from $0.0318 \pm 0.0179$ for forests to $0.176 \pm 0.126$ for grasslands, $RH$ (%) is relative humidity, $LAI$ (m$^2$ m$^{-2}$) is one-sided leaf area index, $\beta = 0.15$ °C$^{-1}$ is a temperature response parameter, and $T$ (°C) is the temperature. Note that the temperature response was originally derived using temperatures scaled to the notional height of trace gas exchange $z_0'$ (m). Since sensible heat flux measurements, which are required for this extrapolation (e.g. Nemitz et al., 2009), were not available for all sites, we here used measured air temperatures instead. The influence of using $T$ and $RH$ at the reference height instead of $z_0'$ is discussed later in this paper. Contrary to the original formulation of Flechard et al. (2010), Massad et al. (2010) do not use absolute values of $|T|$ (°C), but we chose to do so under the assumption that generally $R_{\text{w}}$ increases in freezing conditions (e.g. Erisman and Wyers, 1993).

*(ii) Wichink Kruit et al. (2010)*

Following the bidirectional non-stomatal exchange paradigm introduced in the cuticular capacitance model of Sutton et al. (1998), Wichink Kruit et al. (2010) developed a simplified steady-state non-stomatal compensation point ($\chi_{\text{w}}$) model (henceforth referred to as *WK*) using three years of flux measurements over an unfertilized grassland in the Netherlands. In this model, a simple humidity response after Sutton and Fowler (1993) is used as an approximation for $R_{\text{w}}$ under 'clean conditions':

$$R_{\text{w,WK}} = 2 \cdot e^{\frac{1}{12} \cdot (100 - RH)} \, . \tag{6}$$

$\chi_{\text{w}}$ (µg m$^{-3}$) is calculated from the temperature response of the Henry equilibrium and the ammonium-ammonia dissociation equilibrium, similar to formulations used for the stomatal compensation point (e.g. Nemitz et al. 2000), as:

$$\chi_{\text{w}} = \frac{2.75 \cdot 10^{15}}{T + 273.15} \cdot e^{\left(-\frac{1.04 \cdot 10^4}{T + 273.15}\right)} \cdot \Gamma_{\text{w}} \, , \tag{7}$$

where $\Gamma_{\text{w}}$ (–) is the non-stomatal emission potential and corresponds to the molar ratio of [NH$_4^+$] to [H$^+$] in the leaf surface water layers. Wichink Kruit et al. (2010) derived a functional relationship for $\Gamma_{\text{w}}$ from measurements of the ammonia air concentration at a reference height of 4 m:

$$\Gamma_{\text{w}} = 1.84 \cdot 10^3 \cdot \chi_{\text{a}}\{4\,m\} \cdot e^{-0.11 \cdot T} - 850 \, , \tag{8}$$

The WK model is only structurally bidirectional in that the effect of the pollution climate is shifted from $R_{\text{w}}$ to $\chi_{\text{w}}$. In practice, as $\chi_{\text{w}}$ is parameterized as a fraction of $\chi_{\text{a}}$, no emissions can occur.

An effective non-stomatal resistance, $R_{\text{w,eff.}}$ (s m$^{-1}$), that produces identical results when used with a unidirectional non-stomatal resistance-only model, can be written as:

$$R_{\text{w,eff.}} = \frac{\chi_{\text{c}} \cdot R_{\text{w}}}{\chi_{\text{c}} - \chi_{\text{w}}} \, , \tag{9}$$

or during nighttime conditions, when $R_{\text{s}}$ is here assumed to approach infinity:



$$R_{w,\text{eff.,nighttime}} = \frac{\chi_a\{z-d\}\cdot R_w + \chi_w\cdot(R_a\{z-d\}+R_b)}{\chi_a\{z-d\}-\chi_w}\ . \tag{10}$$

Note that Wichink Kruit et al. (2010) used surface temperatures estimated from outgoing longwave radiation and the Stefan-Boltzmann law, but in practice the model is routinely run with air temperatures within the DEPAC3.11 code (van Zanten et al., 2010). As with the MNS model, the difference between using air and surface temperatures when the latter was available was investigated in a small sensitivity study.

## 2.3 Theoretical considerations and generation of hypotheses

The MNS model uses a minimum non-stomatal resistance $R_{w,\text{min}}$ of 31.5 s m$^{-1}$, which is further significantly increased when $AR < 1$, $RH < 100$ %, $LAI < 1$ and $T \neq 0$ °C (Fig. 2). For example, at $AR = 0.5$ and $T = 10$ °C, the minimum allowed $R_w$ at 100 % relative humidity lies between 163 and 282 s m$^{-1}$ for an $LAI$ range of 1 to 3 m$^2$ m$^{-2}$. It is evident from Tab. 1 of Massad et al. (2010) that $AR < 1$ is no rare occurrence, but compared to minimum measured $R_w$ (ibid.) predicted values appear to be rather high. It should also be noted that in the MNS model, the deposition velocity can never approach the maximum limit allowed by turbulence $v_{d,\text{max}}\{z-d\}$ (m s$^{-1}$):

$$v_{d,\text{max}}\{z-d\} = (R_a\{z-d\}+R_b)^{-1}\ . \tag{11}$$

The temperature dependent parameterization of $\Gamma_w$ in the WK model can lead to contrasting effects: When temperatures increase, the exponential decay function in Eq. (8) can completely counter the growth of Eq. (7). In other words, depending on NH$_3$ air concentration levels, after a certain cut-off temperature the non-stomatal compensation $\chi_w$ point decreases (Fig. 2), although with a constant $\Gamma_w$, an equilibrium shift towards gaseous ammonia would be expected to lead to a further exponential increase of $\chi_w$. Consequently, when $T$ is high and $\chi_w$ approaches zero, $\chi_c$ is canceled out in Eq. (9) and $R_{w,\text{eff.}}$ becomes equal to the clean air $R_{w,\text{WK}}$ (Eq. (6)), which at 100 % relative humidity is as low as 2 s m$^{-1}$.

Based on these considerations, we hypothesize that:

(i) The MNS model has a tendency to overestimate $R_w$ and consequently to underestimate $F_t$, especially at sites with moderately low acid ratios.

(ii) The WK model has a tendency to underestimate $R_w$ and consequently to overestimate $F_t$, especially during moderately high temperatures and low air ammonia concentrations.

## 2.4 Derivation of night-time non-stomatal resistances from flux measurements

Non-stomatal resistance models are parameterized using flux measurements during reasonably turbulent, i.e. near neutral or only slightly stable, nighttime conditions. When stomatal closure is high and therefore $R_s \gg R_w$, we can assume that the canopy resistance $R_c$ is approximately equal to $R_w$ based on the single-layer model when the non-stomatal pathway is treated unidirectional:



$$R_{w,obs.} \approx -\frac{\chi_a\{z-d\}}{F_t} - (R_a\{z-d\} + R_b) \, , \tag{12}$$

where $R_{w,obs.}$ (s m$^{-1}$) is the observed non-stomatal resistance, and $F_t$ is in µg m$^{-2}$ s$^{-1}$. $R_{w,obs.}$ values were selected from turbulent nighttime conditions (e.g. Wichink Kruit et al., 2010), when $R_a\{z-d\} + R_b < 200$ s m$^{-1}$, $u_* > 0.1$ m s$^{-1}$, and global radiation < 10 W m$^{-2}$.

Existing datasets of flux measurements were used for a comparison of measured and modeled $R_w$. These measurements were conducted at two peatland sites, Auchencorth Moss (AM) in the United Kingdom, and Bourtanger Moor (BM) in Germany, as well as three grassland sites, Oensingen (OE) in Switzerland, and Solleveld (SV) and Veenkampen (VK), both in the Netherlands. At AM, OE, SV and VK, the aerodynamic gradient and at BM the eddy covariance method was used. For detailed site and measurement setup descriptions, the reader is referred to Flechard et al. (1999) for AM, Richter et al. (2016)
and Hurkuck et al. (2014) for BM, and Spirig et al. (2010) for OE. SV and VK datasets are unpublished as of now. SV is best characterized as a semi-natural grassland and is located in the dune area west of The Hague, NL. NH$_3$ concentration profiles were measured using a Gradient Ammonia High Accuracy Monitor (GRAHAM, Wichink Kruit et al., 2007) system with inlets at 0.8, 1.7 and 3.6 m above ground. VK is an experimental grassland site used by Wageningen UR for meteorological measurements, where NH$_3$ was sampled at 0.8 and 2.45 m above ground using Differential Optical Absorption Spectroscopy
(DOAS, Volten et al., 2012). A brief overview of measurement conditions at the five sites is given in Tab. 1. $LAI$ and canopy height $h_c$ (m) measurements were available for AM and OE, and the default values proposed in Tab. 6 of Massad et al. (2010) were used at the other sites. Emission events at OE not suitable for this study were filtered out by removing 9 days of measurements after a fertilization events, based on the $e$-folding time of 2.88 days used for fertilizer emission potentials in Massad et al. (2010), which translates into a 95 % 'extinction time' of 8.63 days for the management influence. For VK, no
management logs for the measurement site or the surrounding fields were available and only two strong emission periods were removed manually after visual inspection of the dataset.

## 2.5 Proposal for a semi-dynamic parameterization of non-stomatal emission potentials

The Wichink Kruit et al. (2010) parameterization was developed for frameworks within which the use of dynamic cuticular capacitance models in conjunction leaf surface chemistry modules may not be practical (e.g. to limit computation time of
large scale CTMs). We here additionally investigate the feasibility of a $\Gamma_w$ parameterization based on backward-looking moving averages of air ammonia concentrations as a proxy for prior NH$_3$ inputs into the system which might saturate leaf water layers and enhance the compensation points. If such a relationship exists, it can provide an easy-to-use metric that can be calculated from readily available observations without the need for spinning up and iteratively solving a model for $F_t$ estimates, while still allowing the use of a more mechanistic bidirectional approach to non-stomatal exchange. $\Gamma_w$ values are
derived as done by Wichink Kruit et al. (2010), i.e. $R_w$ is parameterized for clean air according to Eq. (6), $\chi_w$ is calculated as

$$\chi_w = \chi_a\{z-d\} + F_t \cdot \left( R_a\{z-d\} + R_b + R_{w,WK} \right) \, , \tag{13}$$



and finally, $\Gamma_w$ is calculated by rearranging Eq. (7) to:

$$\Gamma_w = \frac{T+273.15}{2.75 \cdot 10^{15}} \cdot e^{\left(\frac{1.04 \cdot 10^4}{T+273.15}\right)} \cdot \chi_w \ . \tag{14}$$

The relationship was investigated for moving-windows of different lengths (1 day, 3 days, 7 days, and 14 days) under exclusion of periods with substantial rainfall ($> 5$ mm d$^{-1}$).

## 3 Results and discussion

### 3.1 Comparison of existing parameterizations with observations

The MNS model tends to underestimate nighttime $F_t$ at all five sites, whereas the WK model overestimates $F_t$ for BM, OE and SV, underestimates it for VK, and only very slightly underestimates it for AM (Fig. 3). Note that total cumulative $F_t$ in Fig. 3 is by no means representative for an estimate for total NH$_3$ input during these times, but based on non-gap filled nighttime fluxes only. Additionally, a mismatch between modeled and measured flux densities early in the time series propagates through the whole time series of cumulative fluxes. For example, at BM the MNS model performs very well after a mismatch during the first week, whereas the WK model fits the observations closely until mid-March 2014. Similarly, the strong measured deposition event early in the VK time series is not reproduced by either of the models. Comparing differences in modeled and measured nighttime $R_w$ (Fig. 4, upper row) supports these observations: While using the MNS model leads to an overestimation of the majority of observed $R_w$ at all sites, as hypothesized, the picture is not as clear for WK. Here, the majority of modeled $R_w$ values lies below the observations for BM, OE, SV and VK, however, for AM and VK both frequent over- and underestimations of $R_w$ canceled each other out, thereby leading to fairly reasonable predicted net fluxes at these two sites. The inverse of these resistances, the non-stomatal conductance $G_w = R_w^{-1}$ may be a better predictor for the resulting fluxes, as very high resistances have a negligible effect on fluxes. Differences between modeled and measured $G_w$ are shown in the lower row of Fig. 4 and generally lead to similar conclusions (note that here underestimations of $G_w$ directly lead to underestimations of $F_t$), but emphasize the relatively good predictive capabilities of MNS at BM and WK at VK during most times, which may not immediately be obvious from looking at cumulative fluxes (Fig. 3).

We attribute the mismatch of the MNS model results and measurements to the relatively high baseline minimum allowed $R_w$ and the strong response of the temperature correction function (Fig. 5, left panel). Note that $AR$ at all sites is lower than 1, ranging from 0.1 at BM to 0.7 at AM, which results in minimum $R_w$ of 315 and 45 s m$^{-1}$ before $LAI$ and $T$ correction, respectively. For example, at OE with an $AR$ of 0.4 and an average $LAI$ of approximately 2 m$^2$ m$^{-2}$, even under conditions highly favoring deposition towards the external leaf surface in the MNS model ($RH = 100$ %, $T = 0$ °C), deposition velocity is restricted to an upper bound of 1.8 cm s$^{-1}$, although observations regularly exceeded this threshold. In their comprehensive literature review, Massad et al. (2010) themselves report $R_{w,min}$ between 1 and 30 s m$^{-1}$ for grassland and between 0.5 and



24 s m$^{-1}$ for semi-natural ecosystems. In their parameterization of $R_w$, on the other hand, the actual deposition velocity can never approach the theoretical limit allowed by turbulence (Eq. (11)), although this case was regularly observed in the field. This is of course true for all unidirectional $R_w$ parameterizations of the commonly used $R_w = R_{w,min} \cdot e^{a \cdot (100-RH)}$ form, however, in the WK model a small minimum $R_w$ of 2 s m$^{-1}$ allows $v_d$ to approach $v_{d,max}$ closely. Regarding the temperature

correction, the parameter $\beta = 0.15$ °C$^{-1}$ translates into an increase of $R_w$ by a factor of 4.5 with a $T$ increase of 10 K. Equation (7), however, only predicts an increase of the compensation point $\chi_w$ by a factor of approximately 2.8 to 4.1 for a $T$ increase of 10 K, depending on the starting temperature, which translates into a significantly smaller factor for $R_{w,eff.}$ considering the influence of other variables in Eq. (9) and / or Eq. (10). Note, the relatively good agreement with measured fluxes at BM, despite the very low $AR$.

Reasons for strikingly diverse performance of the WK model are not straightforward, but may be explained based on the combined effect of $T$ and $\chi_a$ on the $\Gamma_w$ parameterization, as depicted in Fig. 2. For example, at BM the model performs relatively well until mid-March 2014 (Fig. 3), when measured fluxes decrease, whereas modeled fluxes remain at a similar level and later even increase. This observation corresponds to an increase in both $T$ and $\chi_a$ at the site (cf. Richter et al., 2016), leading to a decrease in effective $R_w$ and therefore an increase in modeled $F_t$. In fact, with all sites pooled into one

combined dataset, two interesting characteristics of the parameterization emerge from a plot of differences in modeled and measured $R_w$ against $\chi_a$ (Fig. 5, right panel): (i) The underestimation of $R_w$ does indeed increase with rising temperatures and $\chi_a$, as hypothesized. (ii) There is an additional tendency to actually overestimate $R_w$ when temperatures are relatively low, which strongly responds to increasing $\chi_a$ and may be an indication of a too high modeled $\Gamma_w$ under these conditions. These two contrasting effects may explain the good agreement of net modeled and measured cumulative fluxes e.g. at AM,

where concentrations were relatively low during most times and both low and high temperatures without extremes were measured.

Nighttime $R_{w,obs.}$ are affected by (i) the uncertainty in the flux measurements, which can be high due to insufficient turbulent mixing, and (ii) uncertainty in modeled $R_a\{z-d\}$ and $R_b$, which results from increasingly high stability corrections ($\Psi_M\left\{\frac{z-d}{L}\right\}$ and $\Psi_H\left\{\frac{z-d}{L}\right\}$) under increasing atmospheric stability, possible inaccuracy of estimated $z_0$ and $d$, and

possible inadequacy of the $R_b$ model for some surfaces. We therefore emphasize that the results of this study are to be interpreted qualitatively and can only reveal overall tendencies in the models' accuracy, not provide a precise quantification of the mismatch between models and measurements. Propagation of these uncertainties through the analysis resulted in some negative values of $R_{w,obs.}$. There are generally two possible reasons for negative canopy resistance values to occur: (i) emission (i.e. positive fluxes), or (ii) 'overfast' deposition ($v_d > v_{d,max}$) that is not compatible with the resistance modeling

framework used here. As a rule of thumb, we set an upper tolerance threshold for $v_d$ of $1.5 \cdot v_{d,max}$, considered to be within the limits of night-time flux measurement uncertainty and representing perfect sink behavior, and consequently set $R_{w,obs.}$ to zero in these cases. Measurements where $v_d > 1.5 \cdot v_{d,max}$ were discarded and assumed to be either resulting from incompatibility with the atmospheric resistance ($R_a\{z-d\}$, $R_b$) model or from measurement error. During emission events,





$R_{w,obs.}$ was set to infinity. Ranges from 2 to 16 % invalid values, 63 to 93 % deposition and 4 to 29 % emission and were observed across the five sites during near-neutral nighttime conditions. The latter especially highlights the importance of further research towards a truly bidirectional paradigm for non-stomatal exchange (i.e. cuticular desorption, ground-based emissions, or emission fluxes from other environmental surfaces).

An additional investigation of daytime non-stomatal exchange would be beneficial in terms of a significant reduction of uncertainty in the observations and in order to cover a much wider range of temperatures and humidity regimes. However, comparisons based on daytime flux estimates were not made in this study in order not to introduce an additional source of bias via the stomatal pathway. Both Massad et al. (2010) and Wichink Kruit et al. (2010) also presented parameterizations for the stomatal emission potential, $\Gamma_s$ (–). However, for MNS information about annual total (dry and wet) N input into the

system is necessary. While this can be estimated from spinning up a model that incorporates more reactive nitrogen species than just $NH_3$, we do not feel confident estimating total N input from modeled $NH_3$ dry deposition alone. At sites where total N input is known (e.g. BM, from Hurkuck et al. (2014), or from CTM results), the MNS and WK parameterizations predict such different $\Gamma_s$ estimates that one model predicts net emission from the stomata and one model predicts a net uptake over the course of the measurement campaign. A detailed investigation on the reasons for this mismatch is, however, beyond the

scope of this paper.

Another source of uncertainty lies in the fact that $R_w$ models are often developed as 'cuticular resistance' models with only leaf surface exchange in mind. However, in the one-layer resistance framework used here it is not possible to clearly differentiate between deposition towards or emission from wet leaf surfaces, leaf litter, the soil, stems and branches, and any other environmental surface. In fact, the MNS model was originally developed on the basis of the two-layer model of Nemitz

et al. (2001), but outside of management events, the ground layer resistance was set to infinity (Massad et al., 2010) and the model reduces to a one-layer model. While it is indeed conceptually unsatisfactory to ignore the source / sink strength of the ground-layer, an unambiguous identification of multiple non-stomatal pathways' flux contributions by simply inverting the model and inferring resistances from meteorological measurements is not possible, unless there is a signal that can confidently be attributed to originate from e.g. the ground layer (for instance after fertilizer application). Therefore, due to

these methodological limitations, both the parameterizations and the measurements of $R_w$ discussed in this paper may very well integrate exchange fluxes with not only wet leaves, but also e.g., the the soil, stems and branches, or other surfaces.

## 3.2 Semi-dynamic $\Gamma_w$

Estimated non-stomatal emission potentials $\Gamma_w$ appear to have a strong dependency on backward-looking moving averages of measured air ammonia concentrations $\chi_{a,mov.avg.}$ (µg m$^{-3}$) (Fig. 6). While this may indicate some potential as an easy-to-use

and readily available proxy for prior $NH_3$ inputs without the need for more complex and / or computationally intensive mechanistic models, estimated $\Gamma_w$ values are extremely noisy and span multiple orders of magnitude in the < 5 µg m$^{-3}$ range. An increase in the moving-window length from 1 day (Fig. 6a) to 14 days (Fig. 6d) does not lead to a substantial decrease in





the magnitude of the noise. There is a very clear linear relationship when log-transforming both $\Gamma_w$ and $\chi_{a,mov.avg.}$ ($R^2 = 0.62$ for the 1 d moving average case; not shown), however, the strong variability of the data, especially in the low-concentration region, leads to a best fit that predicts large $\Gamma_w$ even at concentrations as low as 1 µg m⁻³ ($\Gamma_w \approx 380$), which eventually ends in unreasonably high emission fluxes. Without further noise reduction, this approach appears unfeasible as an alternative to

more sophisticated dynamic models (e.g. Flechard et al., 1999) or those featuring additional dependencies as the one of Wichink Kruit et al. (2010). Making the moving-window width dependent on time since the last substantial precipitation event might help reduce this noise and lead to a more realistic representation, but in turn complicates the implementation and increases the degrees of freedom in this approach, thereby reducing its advantage over mechanistically more accurate models.

### 3.3 MNS with updated parameters

Since we hypothesized the reasons for the mismatch between modeled $R_w$ with the MNS model and measured $R_{w,obs.}$ to be based on two easily accessible parameters with relatively obvious effects on modeled resistances ($R_{w,min}$ and the temperature response parameter $\beta$ in Eq. (5)), we additionally investigated the effects of adjusting them towards smaller values. Figure 7 shows the effects of simply halving both $R_{w,min}$ and $\beta$ on predicted nighttime fluxes. Even though there still

remains significant scatter, doing so decreases the mismatch between modeled and measured fluxes in most cases. However, in one case (BM) predicted fluxes actually turn out to fit the measurements worse than with the original parameters, and in another case (VK) this only leads to a marginal improvement. While there does not appear to be a comprehensive, generic solution, we assume that there is potential for a significant overall improvement by optimizing these two parameters based on independent data from all four ecosystem types (grassland, arable, forest and semi-natural ecosystems) used in this

parameterization.

### 3.4 Sensitivity of the main findings

Parts of both models used in this study were developed using an estimate of surface temperatures, either by extrapolating $T$ from the reference height $z - d$ to the notional height of trace gas exchange $z_0'$ using sensible heat flux $H$ (W m⁻²) measurements, or by estimating $T\{z_0'\}$ from outgoing long wave radiation measurements and the Stefan-Boltzmann law.

Additionally, the temperature response function of Flechard et al. (2010), which is used within the MNS model, was fitted using surface level values of relative humidity, $RH\{z_0'\}$ which were derived using measured latent heat fluxes $LE$ (cf. Nemitz et al., 2009). Since $H$ and $LE$ measurements were not available at all sites and introduce an additional source of uncertainty, especially during moderately stable nighttime conditions, and the WK model is routinely being used with air temperatures within the DEPAC3.11 code, we here used both $T$ and $RH$ at the reference height as input data. Figure 8 (upper

row) illustrates the effects of using $T$ and $RH$ at different conceptual model heights for AM. While there are of course





numerical differences, the impact on this study's main findings are negligible. Generally, the WK model appears to be less sensitive to these choices than the MNS model.

For both SV and VK, no measurements of [HNO$_3$] and [HCl] were available. We estimated $AR$ for the MNS model based on the observations of Fowler et al. (2009), that across NitroEurope sites, [SO$_2$] makes up around 40 % of the sum

[SO$_2$]+[HNO$_3$]+[HCl] to be approximately 3.5 times the ratio of [SO$_2$]/[NH$_3$]. From the definitions $AR$ = (2[SO$_2$]+[HCl]+[HNO$_3$])/[NH$_3$] and $SN$ = [SO$_2$]/[NH$_3$], a lower bound of $AR \geq 2 \cdot SN$ is obvious. Using a symmetrical range around our initial estimate of $AR \approx 3.5 \cdot SN$, we set an additional upper bound of $AR \leq 5 \cdot SN$ and tested the effects of using these values on $R_w$ differences for both affected sites (Fig. 8, lower row). Again, there are apparent numerical differences, but they do not affect the main observations made here (i.e. they neither change the sign of the differences in

modeled and measured $R_w$, nor do they change the general magnitude of the differences e.g. from a strong overestimation to an insignificant one).

## 4 Conclusions and recommendations for further research

We presented a semi-quantitative assessment of the compared performances of two state-of-the-art non-stomatal resistance parameterizations for ammonia biosphere-atmosphere exchange models, supported by flux measurements from two semi-

natural peatland and three grassland sites.

The unidirectional $R_w$-only approach of Massad et al. (2010), which, in addition to the classical humidity response, reflects the effects of the pollution climate, vegetation via the leaf area index, and an empirical temperature response, was found to overestimate $R_w$ during nighttime at all five sites. We tested the potential for an easily accessible improvement of predicted $R_w$ and consequently predicted NH$_3$ exchange fluxes by using smaller values for the temperature response and minimum $R_w$

parameters and propose to further investigate this route using data from all four ecosystem types represented in the MNS $R_w$ parameterization.

The quasi-bidirectional model of Wichink Kruit et al. (2010) shows a more complex response to varying pollution climates and meteorological conditions, with both a tendency to underestimate $R_w$, as initially hypothesized, during warm conditions and moderately high ambient NH$_3$ concentrations, and a tendency to overestimate $R_w$ during colder conditions, with an even

stronger response to increasing $\chi_a$. While there is likely no simple solution as may be the case for the MNS model, the WK parameterization with its non-stomatal compensation point approach appears to be conceptually more compatible with field observations (e.g. morning peaks of NH$_3$ emission due to evaporation of leaf surface water). We strongly encourage revisiting the $\Gamma_w$ parameterization with additional data from other ecosystems and investigating alternative approaches to model the effects of seasonality in $\Gamma_w$, e.g. by using a smoothed temperature response instead of an instantaneous one. An

extension of the model with an SO$_2$ co-deposition response is currently being researched.

A simple alternative approach to dynamic models for the non-stomatal emission potential revealed a clear response of $\Gamma_w$ to backward-looking moving averages of $\chi_a$. These findings may turn out to be promising for CTMs, as they provide a first



step towards a simplification of computationally intensive mechanistic model. However, further noise reduction, especially in the low concentration region, is needed for it to be useful for predicting $NH_3$ exchange fluxes.

## Acknowledgements

We greatly acknowledge funding of this work by the German Federal Ministry of Education and Research (BMBF) within the junior research group NITROSPHERE under support code FKZ 01LN1308A. The authors are grateful to all scientific and technical staff involved in gathering the data used in this study. Many thanks to R.-S. Massad for her helpful comments and clarifications during the early stages of developing the program code used for the flux calculations. Finally, we thank C. Ammann for his valuable comments on the manuscript and his contribution to the OE dataset.

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





**Tables**

**Table 1: Summary of the five datasets. AGM = Aerodynamic gradient method; EC = Eddy covariance, MNS = Massad et al. (2010). Measurement period is the period during which flux measurement were available after final data filtering. $T$ and $\chi_a$ ranges are minimum and maximum values during the measurement period and values in parentheses denote the 5 %, 50 %, and 95 % quantiles.**

| ID | Site name | Ecosystem type in MNS | Measurement period | Measurement technique | $T$ (°C) | $\chi_a$ (µg m-3) | avg. $AR$ (–) | Reference |
|----|-----------|----------------------|--------------------|----------------------|----------|------------------|---------------|-----------|
| AM | Auchencorth Moss (UK) | semi-natural | 02/95 – 02/96 05/98 – 11/98 | AGM | -7.8 – 26.9 (0.0, 9.4, 17.3) | 0.0 – 32.9 (0.1, 0.4, 2.9) | 0.7 | Flechard et al. (1999) |
| BM | Bourtanger Moor (DE) | semi-natural | 02/14 – 05/14 | EC | -4.4 – 22.3 (0.7, 7.3, 17.8) | 1.6 – 62.0 (3.2, 9.0, 26.6) | 0.1 | Richter et al. (2016) |
| OE | Oensingen (CH) | grassland | 07/06 – 10/07 | AGM | -3.0 – 33.1 (1.2, 12.3, 23.8) | 0.0 – 24.7 (0.4, 2.2, 8.0) | 0.4 | Spirig et al. (2010) |
| SV | Solleveld (NL) | grassland | 09/14 – 08/15 | AGM | -1.5 – 31.7 (3.4, 11.6, 20.4) | 0.1 – 15.6 (0.2, 1.2, 6.6) | 0.5 | unpublished |
| VK | Veenkampen (NL) | grassland | 01/12 – 10/13 | AGM | -5.4 – 31.6 (4.0, 15.2, 26.2) | 0.3 – 116.9 (2.5, 8.8, 27.7) | 0.3 | unpublished |



**Figures**

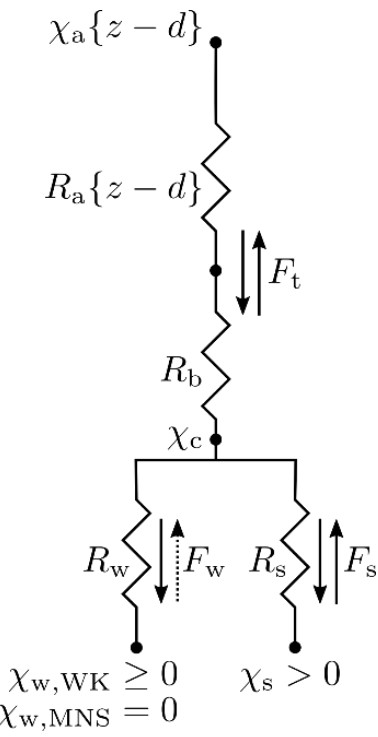

**Figure 1: Structure of the single-layer model of NH₃ surface-atmosphere exchange used in this study. The non-stomatal pathway can be treated either uni- or bidirectionally, depending on the specific parameterization. MNS = Massad et al. (2010); WK = (Wichink Kruit et al., 2010).**





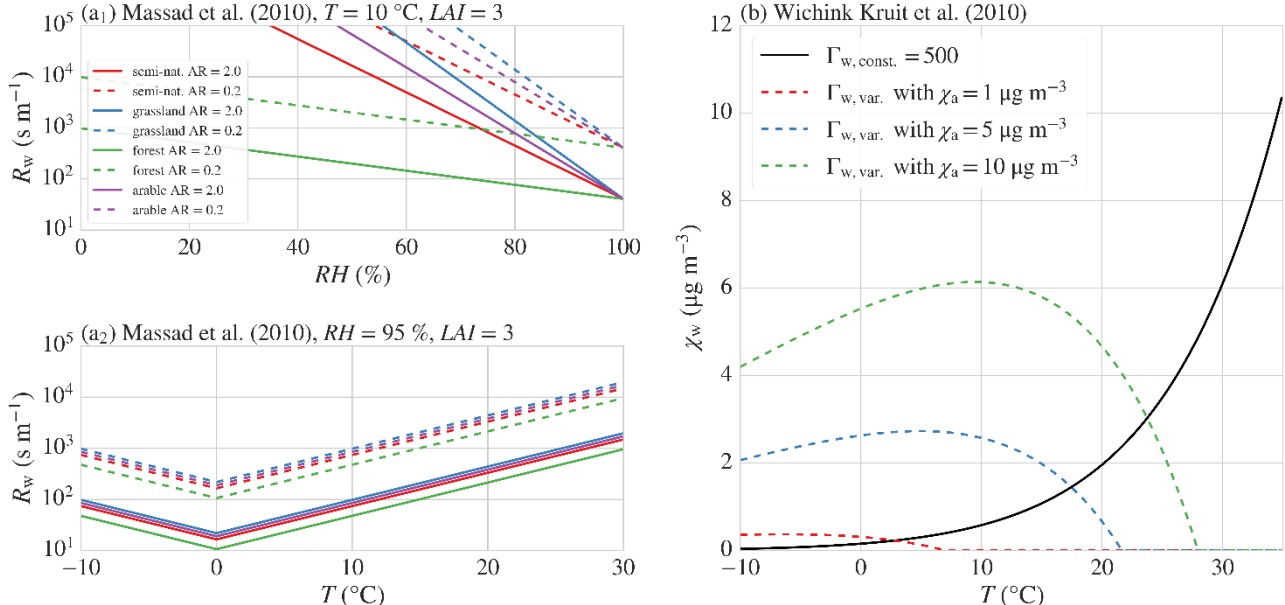

**Figure 2: Theoretical considerations about the non-stomatal resistance parameterizations' response to changes in micrometeorological conditions. (a) Non-stomatal resistance ($R_w$) as a function of (a$_1$) relative humidity ($RH$) and (a$_2$) temperature ($T$) for different ecosystems and pollution climates according to the Massad et al. (2010) parameterization. (b) Non-stomatal compensation point ($\chi_w$) as a function of air ammonia concentration ($\chi_a$) and temperature ($T$) in the Wichink Kruit et al. (2010) parameterization.**

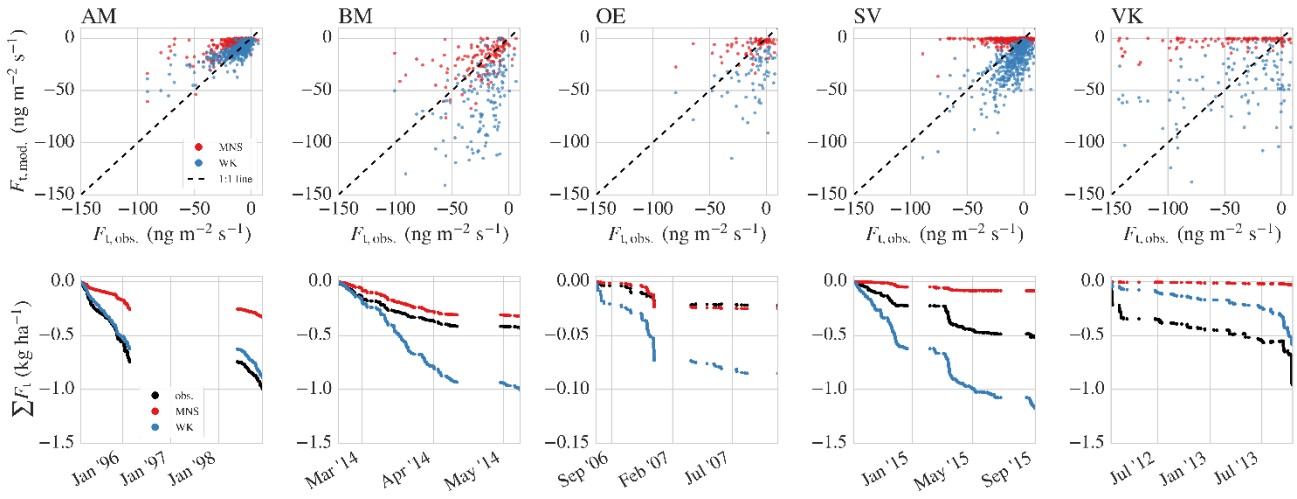

**Figure 3: Measured and modeled ammonia dry deposition fluxes ($F_t$) during near-neutral or slightly stable nighttime conditions. Upper row: Modeled vs. measured 6 h median flux densities. Lower row: Cumulative fluxes. obs. = observations; MNS = Massad et al. (2010); WK = Wichink Kruit et al. (2010). Refer to the text for site descriptors. Note the different scaling of the axes.**





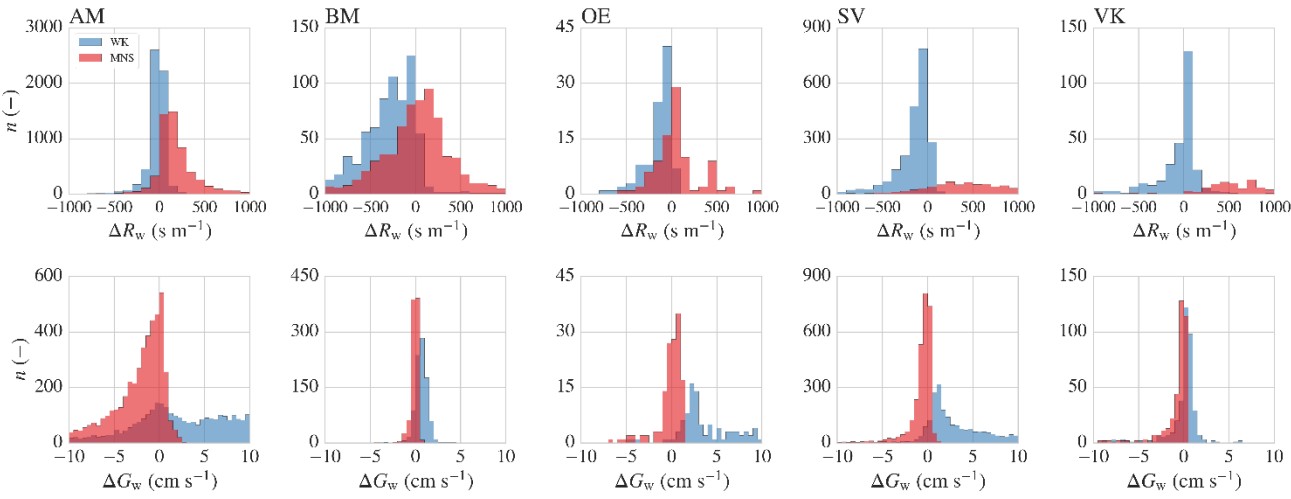

**Figure 4: Differences in measured and modeled 30 min nighttime non-stomatal resistances ($R_w$, upper row, 100 s m$^{-1}$ bins) and conductances ($G_w$, lower row, 0.5 cm s$^{-1}$ bins). $\Delta R_w = R_{w,modeled} - R_{w,observed}$ and $\Delta G_w = G_{w,modeled} - G_{w,observed}$, i.e. positive values indicate an overestimation and negative values indicate an underestimation by the models. Note that an overestimation of $R_w$ leads to an underestimation of fluxes $F_t$, whereas an overestimation of $G_w$ leads to an overestimation of $F_t$.**

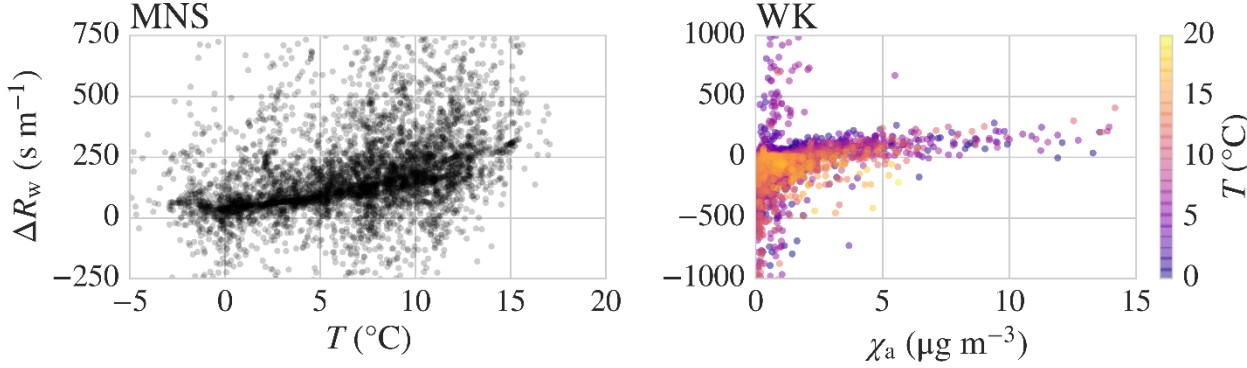

**Figure 5: Differences between modeled and measured 30 min nighttime non-stomatal resistances ($\Delta R_w$) as a function of $T$ and/or $\chi_a$. Left panel: Increasing mismatch of measured and modeled $R_w$ in the MNS model due to a too strong $T$ response. The line-shaped pattern emerges from times when observed $R_w$ is zero and is equal in magnitude to the minimum allowed $R_w$ in the parameterization. Right panel: The WK model reveals a tendency for both stronger over- and underestimation of observed $R_w$ with increasing $\chi_a$, where overestimation occurs more frequently during colder and underestimation during warmer conditions.**





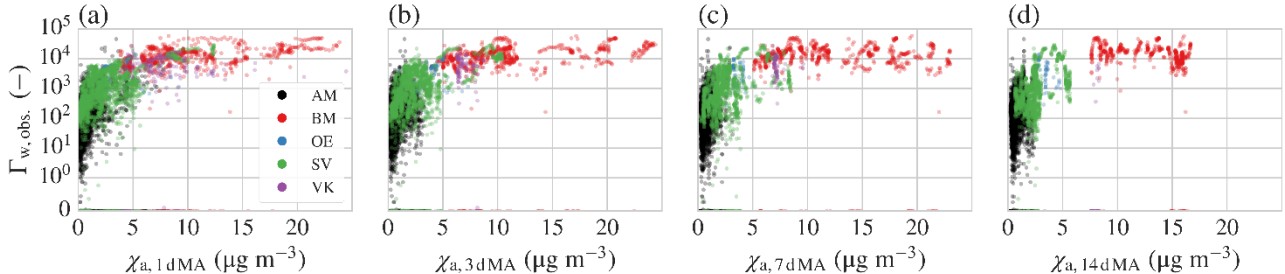

**Figure 6: Non-stomatal emission potentials inferred from measurements ($\Gamma_w$) as a function of backward-looking moving averages of measured air ammonia concentrations ($\chi_a$). (a) 1 day, (b) 3 day, (c) 7 day, (d) 14 day moving window. Periods with substantial precipitation were removed from the analysis.**

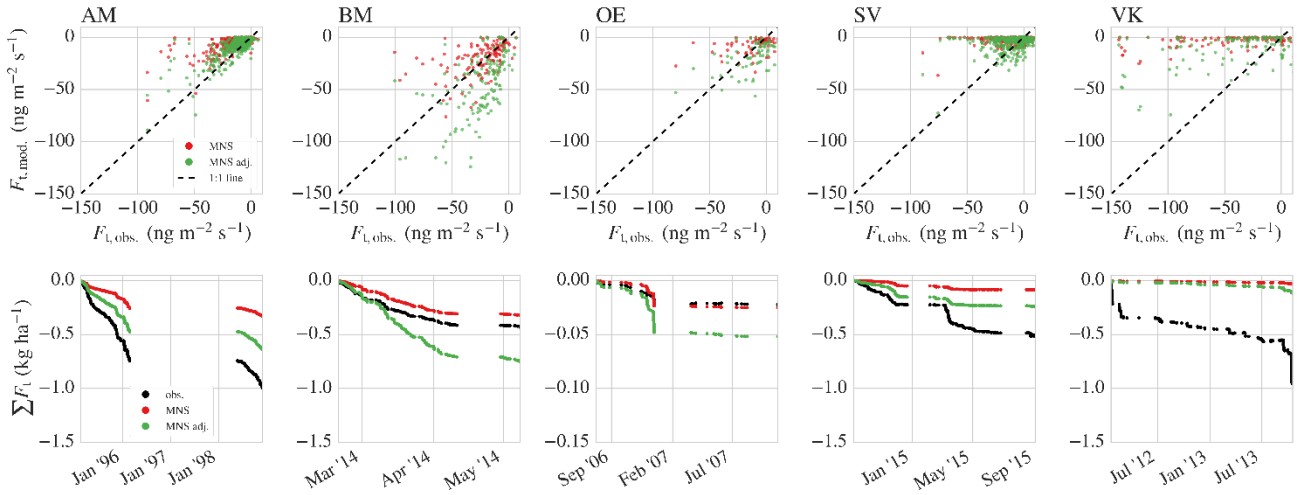

**Figure 7: Measured and modeled ammonia dry deposition fluxes ($F_t$) during near-neutral or slightly stable nighttime conditions. Upper row: Modeled vs. measured 6 h median flux densities. Lower row: Cumulative fluxes. MNS adj. = MNS with halved minimum $R_w$ and temperature response parameter $\beta$.**





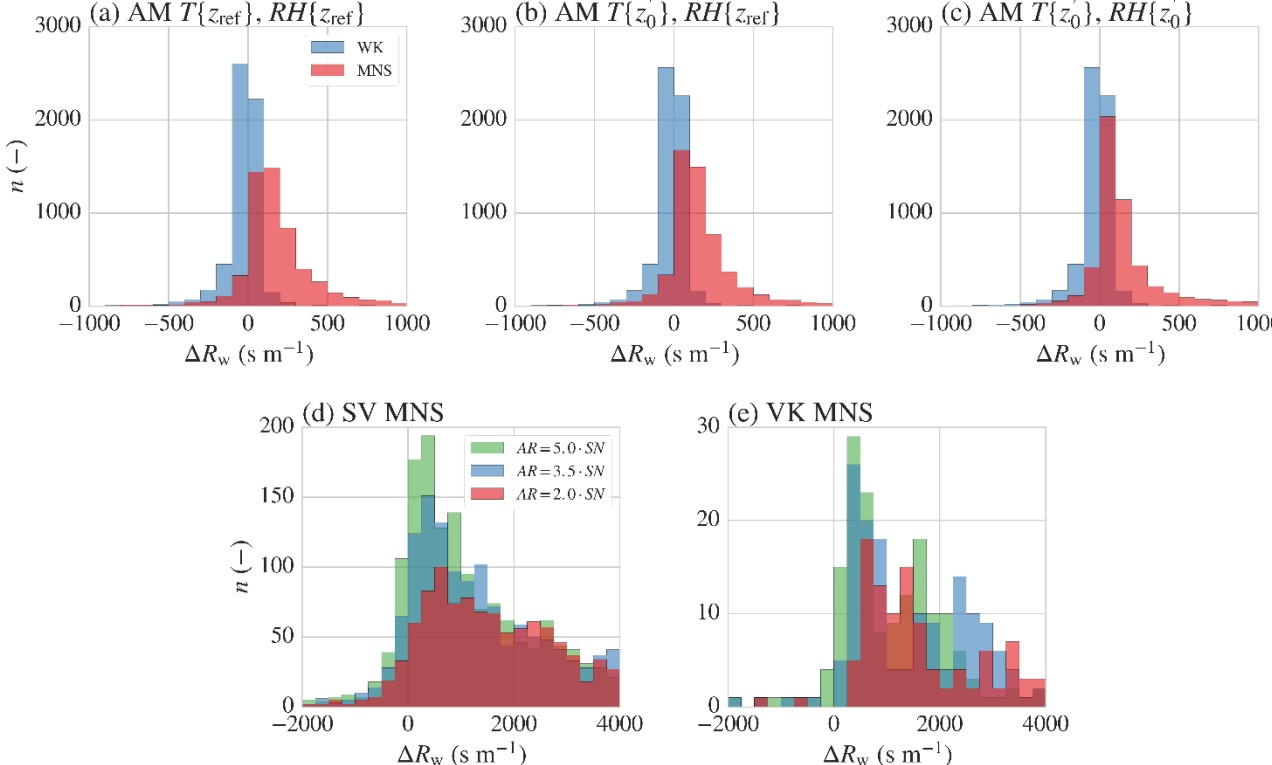

**Figure 8: Sensitivity of differences in measured and modeled non-stomatal resistances to the use of measured air vs. surface temperature and relative humidity estimates. Upper row: Exemplary calculations for AM with (a) $T$ and $RH$ at the reference height, (b) $T$ at the notional height of trace gas exchange ($z_0'$), and (c) $T$ and $RH$ at $z_0'$. Lower row: AR estimated as 2.0, 3.5 and 5.0 times the $[SO_2]/[NH_3]$ ratio SN for (d) Solleveld and (e) Veenkampen. Note the asymmetric horizontal axis in (d) and (e). Data are binned into 100 s m$^{-1}$ bins for (a-c) and 250 s m$^{-1}$ bins for (d-e) to ensure visual clarity.**