# Peer review of "Non-stomatal exchange in ammonia dry deposition models: Comparison of two state-of-the-art approaches"

_Atmospheric Chemistry and Physics, 2016_

## Referee Comment (RC1) · Anonymous Referee #2 · 8 Jul 2016

Review on manuscript acp-2016-403. Non-stomatal exchange in ammonia dry deposition models: Comparison of two state-of-the-art approaches By Frederik Schrader, Christian Brümmer, Chris R. Flechard, Roy J. Wichink Kruit , Margreet C. van Zanten, Undine Richter, Arjan Hensen, Jan Willem Erisman. . .

The topic of the paper is about the comparison of two state of the art approaches for modeling non stomatal exchange in ammonia dry deposition. Several sensitivity tests have been performed to understand the role of biophysical parameters, such as temperature and concentration of ammonia, in five field sites in Europe.

General comments

The paper is within the scope of ACP.. The results are correctly presented; the figures illustrate the results in a clear way, but the order should be changed, as detailed in the specific comments. The paper is written in good English. I recommend this paper to be published in ACP, after some major corrections and improvements in the presentation of results. My main remarks concern principally the way results are presented. In my opinion the results could be presented in a more positive way. The reader cannot be convinced if the authors present their results without highlighting the advantages found after the sensitivity tests. This is detailed in the specific comments. Some bibliography about how these two models have been used until this study would have been necessary to help the reader understand where the authors want to go and why they have chosen these particular models and not others. Did these models give satisfying results in other studies and why did the authors choose them. Partial conclusions at the end of each paragraph need to be more clearly assessed. The go home message needs a clearer explaination.

Specific comments

Abstract. The abstract gives a clear idea of what is presented in the paper. The sentence line 25 page 1 "The proposed $\Gamma$w parameterization..." needs to be detailed to let the reader know in what way it can be improved.

Page 3 line 13: could you explain how it is realistic or not to switch off the soil/leaf litter layer for natural ecosystems, where it can be an important source of NH3, such as mentioned for example in Wentworth et al., 2014.

Page 5 line 15: please give the NH3 concentration under which clean conditions are considered.

Page 5 line 24: this term of "pollution climate" is difficult to understand because it is not precise enough. Do you mean "air pollution climate"as mentioned in Wichink Kruit et al. 2007? Is there a value for NH3 concentration to define this threshold of pollution?

Page 7 line 24 add "with" between "conjunction" and "leaf".

Page 9. The "results and discussion" paragraph needs to be restructured. Uncertainties should be discussed in a specific sub-paragraph. It would be interesting to specify the conditions where these models have been applied, how successful they were, and where they cannot be applied, for example when emission occur instead of deposition.

Page 10 line 1: remove "and" at the end of the line.

Page 10 line 10: "a model": what model exactly are you talking about?

Page 10 lines 10 to 15: This explanation is not clear. These lines have to be rewritten. Line 11, after the sentence "we do not feel confident...", is it supposing that only NH3 dry deposition is available? Line 14-15: "A detailed description..." if the investigation is beyond the scope of the paper why then talking about it and give the results of the sensitivity test if you do not give the reasons of why it could not work? Some ideas could be provided to help the reader understand.

Page 10 line 24-25: What do you mean by "very well"? Do you mean that the assumption of ground layer resistance = infinite is not realistic? And what about weak ground resistance and infinite stomatal resistance? The authors should give some more explanations and overall extract the main positive idea of such sensitivity tests explained in this paragraph. The reader is a bit frustrated not to know if good ideas have to be extracted from that.

Page 10 line 28: The reader cannot understand the ideas mentioned in this 3.2 paragraph if the authors do not explain in what purpose they use moving averages of NH3 concentrations. What is the goal of this exercise?

Page 11 line 2: why this case is not shown? It would have been interesting to see the results?

Page 11 line 9 :The authors give indications of potential improvements and conclude by writing that no improvement is deduced. What is then the purpose of giving these

results if they do not lead to improvement? It should be far better to highlight the advantages instead of giving the disadvantages.

Page 11 line 20: same remark as above. The way this paragraph is written does not give a positive issue. The authors should turn it differently to highlight the positive points. This part should follow figure 3.

Page 12 line 1: What do you mean by "the impact of this study's main findings are negligible"?

Page 12 line 12: Again what is the advantage of doing this if no solution is going out?

Page 12 line 13: the title is not appropriated. Should be "conclusions".

Page 12 line 17: "pollution climate" is not an understandable term. Conclusion needs to be more striking.

Page 12 line 27: "We strongly encourage" is not appropriate. Please reformulate.

Changes in the structure are needed. Figure 7 should follow figure 3, figure 8 should follow figure 4. Please adapt the text in function of these figure changes.

Technical corrections

Page 6 line 10 and line 26, ibid and i.e. have to be in italics. Throughout the text latin expressions should be in italics.

---

## Referee Comment (RC2) · Anonymous Referee #1 · 20 Aug 2016

The paper compares two alternative models to simulate ammonia fluxes and their comparison with the measured fluxes in five peatland and grassland sites, focusing on the non-stomatal fluxes. The motivation for such more empirical deposition models is the inclusion of bi-directional ammonia fluxes into chemical transport models, which requires few and easily available parameters for the surface resistance. The improvement of such models is needed and the respective analysis here is valuable, specifically as it includes a suffiecient number of sites. Focusing exclusively on nighttime fluxes with sufficiently turbulent condition is a good approach. It should, however, be discussed, if nocturnal transpiration could have confounded these observations. The parameters included in both models are temperature and, importantly, relative humid-

ity (RH), whereas different ways are chosen regarding the fate of depositing ammonia, either unidirectional (MNS model) or 'quasi-bidirectional' (WK model). Based on common patterns of the five test sites, systematic under- and overestimation of fluxes are then diagnosed and empirical improvements are suggested by the authors. Although ultimately I agree with the overall direction of the paper and the interpretation of the results (see few exceptions below), I find it difficult to follow. One of the reasons is the continuous introduction of a multitude of parameters which makes consequent reading sometimes time-consuming and frustrating. Even in case it does not agree with the usual policy of this journal, I therefore suggest a table detailing all used parameters with units and possibly also other abbreviations. A second and somewhat related reason is the initial explanation of the two models which in some important places is not sufficiently detailed – some examples are given below. I wonder why the effect of RH is so little discussed in the paper - it has an exponential influence on Rw and thus is the most important independent parameter. It could e.g. be included in a analysis similar to Fig. 5. I also wonder if the effect of backward-looking moving averages shouldn't be evaluated together with the RH history. Saturation effects (as mentioned in p.2, l. 20) could play a role at low RH.

P. 5, l. 24/25: This is difficult to follow. Can it be supported by a formula? What happens if RH decreases?

P. 8, l. 19-23: This is indeed intriguing, but on the other hand I cannot really believe that MNS works so well in the prediction of fluxes at VK, when looking at the cumulative fluxes in Fig. 3. Even during the flat part, there is an underestimation of 0.3 kg ha-1. The shape of the cumulative fluxes at BM is considerably different from VK, while the shapes of ïĄĎGw differences in Fig. 4 are very similar between BM and VK. Please check if the statement is really correct.

P. 11, l. 19: which parameterization?

Figure 2: Why is Rw lowest at T=0°C?

Figure 4: Upper row: There seems to be a mismatch between the number of binned values used for MNS and WK comparison, at least for VK. What is the reason?

Minor issues P.6, l.11: 'approach', better: 'reach'? (also p. 9, l. 2) P. 6, l. 16: 'compensation point Xw decreases' P. 6, l. 23: why 'moderately'? I would suggest to omit this word P. 7, l. 18: event P. 10, l. 1: omit 'and'

---

## Author Comment (AC1) · 12 Oct 2016

**Reply to Anonymous Referee #2**

**General comments**

The paper is within the scope of ACP.. The results are correctly presented; the figures illustrate the results in a clear way, but the order should be changed, as detailed in the specific comments. The paper is written in good English. I recommend this paper to be published in ACP, after some major corrections and improvements in the presentation of results. My main remarks concern principally the way results are presented. In my opinion the results could be presented in a more positive way. The reader cannot be convinced if the authors present their results without highlighting the advantages found after the sensitivity tests. This is detailed in the specific comments.

Thank you for your insightful review. Please refer to the specific comments for a detailed answer to each of your concerns.

Some bibliography about how these two models have been used until this study would have been necessary to help the reader understand where the authors want to go and why they have chosen these particular models and not others. Did these models give satisfying results in other studies and why did the authors choose them.

Thank you for highlighting that we missed to give a justification for the choice of the parameterizations. This has been added to the introduction section. There are indeed notable other models, such as the one of Zhang et al. (2010, 2003). The choice of the two particular models compared in this study is based on a number of different reasons: a) they are structurally very similar (the WK parameterization is flexible in terms of its usage within a one- or two-layer model); whereas Massad et al. (2010) and Zhang et al. (2010) exhibit some fundamental differences in their handling of the ground-layer pathway (this is not discussed in particular in this study, but it leads to difficulties in programming comparable model codes); b) the motivation for this study arose from prior experience of the lead author with the MNS and WK parameterizations; and c) the WK parameterization features a unique handling of the non-stomatal pathway with its 'quasi-bidirectionality', and we found it interesting to see how it compares to the traditional deposition-only approach.

**Changes to the manuscript:** P2L31: Add "The Massad et al. (2010) parameterization has received widespread acceptance in the community, with 53 citations according to the literature database 'Thomson Reuters Web of Science' at the time of writing this article, and variants of it have been applied in numerous studies, e.g. recently in Shen et al. (2016), Móring et al. (2016), Zöll et al. (2016), and others. Wichink Kruit et al. (2010) followed a unique approach by simplifying complex dynamic approaches towards an empirical steady-state formulation of a non-stomatal compensation point model, which is nowadays used within the DEPAC3.11 deposition module (van Zanten et al., 2010) and the chemistry transport model LOTOS-EUROS (Wichink Kruit et al., 2012), and it is structurally compatible with the Massad et al. (2010) model."; Add references for Shen et al. (2016), Móring et al. (2016), Wichink Kruit et al. (2012).

Partial conclusions at the end of each paragraph need to be more clearly assessed. The go home message needs a clearer explaination.

We agree, thank you for pointing out that the conclusions are not clear enough. Please refer to our answers to specific remarks below.

**Changes to the manuscript:** See below.

**Specific comments**

Abstract. The abstract gives a clear idea of what is presented in the paper. The sentence line 25 page 1 "The proposed $\Gamma w$ parameterization..." needs to be detailed to let the reader know in what way it can be improved.

Agreed, this will be more detailed in the revised manuscript.

**Changes to the manuscript:** P1L25-26: Replace "The proposed $\Gamma_w$ parameterization appears to have potential for improvement, but cannot be recommended for use in large scale simulations in its present state due to large uncertainties." with "The proposed $\Gamma_w$ parameterization revealed a clear functional relationship between backward-looking moving averages of air $NH_3$ concentrations and non-stomatal emission potentials, but further reduction of uncertainty is needed for it to be useful across different sites."

Page 3 line 13: could you explain how it is realistic or not to switch off the soil/leaf litter layer for natural ecosystems, where it can be an important source of NH3, such as mentioned for example in Wentworth et al., 2014.

Thank you for hinting at Wentworth et al. (2014). We agree with the reviewer about the importance of soil-based emissions.

The decision to use a single-layer framework for unmanaged ecosystems by Massad et al. (2010) was less based on natural conditions, but primarily on data availability of $\Gamma_g$ for unmanaged ecosystems at the time, as well as methodological issues: Most of the compensation point measurements that they used were based on micrometeorological measurements. The flux measurements used to derive major parts of the parameterization were representative for the ecosystem scale, and the attribution to different conceptual compartments (stomata, cuticula, ground-layer) had to be made based on inverting the resistance model for different environmental conditions (humidity, radiation/time of day). However, this can only be done easily in a single-layer framework due to the relatively straightforward differentiation between stomatal and non-stomatal contributions to measured fluxes (note that we have added discussion on nocturnal stomatal fluxes as suggested by Anonymous Referee #1). Adding a third (ground-layer) pathway severely complicates this approach unless the ground-layer based signal completely dominates the observed fluxes. Ground-layer based emissions are not being ignored, but rather integrated into the stomatal emission potential for unmanaged ecosystems in the MNS parameterization. The model switches to a two-layer formulation after management events, when the contribution of these emissions is strong enough to be more or less unambiguously attributed to the soil. We agree that this is conceptually unsatisfying und should be improved upon in future developments.

Note that this is very similar to our reasoning why we chose to call $R_w$ "non-stomatal" instead of the often used "cuticular" or "external leaf surface" resistance – we cannot be 100 % certain that we do not integrate the influence of other surfaces when we simply invert the model for nighttime conditions (in fact, we most likely do).

**Changes to the manuscript:** See below (Answer to P10L24-25).

Page 5 line 15: please give the NH3 concentration under which clean conditions are considered.

This goes back to Milford et al. (2001), who concluded that Eq. (6) with a minimum $R_w$ parameter of 2 and an exponential decay constant of 1/12 is valid for conditions without $NH_3$ saturation at the leaf cuticles (although the term 'clean conditions' was introduced by Wichink Kruit et al. (2010)).

**Changes to the manuscript: P5L15:** Replace "In this model, a simple humidity response after Sutton and Fowler (1993) is used as an approximation for $R_w$ under 'clean conditions':" with "In this model, a simple humidity response after Sutton and Fowler (1993) is used as an approximation for $R_w$ under low ambient $NH_3$ concentrations, where saturation of the external leaf surfaces is unlikely (Wichink Kruit et al., 2010; Milford et al., 2001):". Add Milford et al. (2001) to the references.

Page 5 line 24: this term of "pollution climate" is difficult to understand because it is not precise enough. Do you mean "air pollution climate"as mentioned in Wichink Kruit et al. 2007? Is there a value for NH3 concentration to define this threshold of pollution?

Yes, we will change this to "air pollution climate". There is no threshold for "pollution" here, rather the (admittedly somewhat vague) term is often used throughout the literature to describe the composition of ambient air in terms of different airborne pollutants.

**Changes to the manuscript:** P4L21, P5L24, P12L17, P12L22: Add "air" before "pollution climate(s)".

Page 7 line 24 add "with" between "conjunction" and "leaf".

Thanks, corrected.

**Changes to the manuscript:** P7L24: Add "with" before "leaf".

Page 9. The "results and discussion" paragraph needs to be restructured. Uncertainties should be discussed in a specific sub-paragraph. It would be interesting to specify the conditions where these models have been applied, how successful they were, and where they cannot be applied, for example when emission occur instead of deposition.

We agree with the reviewer and have added a sub-paragraph in which we discuss the uncertainties of our methods. Regarding the applicability of the models, please refer to the answer to one comment above (about why we chose these particular parameterizations).

**Changes to the manuscript:** Move P9L22-P10L26 and related discussion added during the revision process into new sub-section 3.5 "Sources of uncertainty" at the end of the "Results" section.

Page 10 line 1: remove "and" at the end of the line.

Done, thanks.

**Changes to the manuscript:** P10L1: Remove second "and" in the line.

Page 10 line 10: "a model": what model exactly are you talking about?

No specific model; N-Input could be derived from any kind of model that is able to predict net annual reactive nitrogen deposition.

**Changes to the manuscript:** See below.

Page 10 lines 10 to 15: This explanation is not clear. These lines have to be rewritten. Line 11, after the sentence "we do not feel confident...", is it supposing that only NH3 dry deposition is available? Line 14-15: "A detailed description..." if the investigation is beyond the scope of the paper why then talking about it and give the results of the sensitivity test if you do not give the reasons of why it could not work? Some ideas could be provided to help the reader understand.

Apologies for being unclear here. L11: Yes. L14-15: This statement was primarily given as a justification for why we did not incorporate daytime data by modeling the stomatal pathway, although the flux measurements are less prone to error due to better turbulent mixing during daytime. We believe that this could be a good idea for further studies, where input data for mechanistically satisfying (e.g. photosynthesis-based) models for $R_s$ and reliable estimates of the stomatal emission potential (e.g. via bioassays) are available.

**Changes to the manuscript:** Rewrite P10L10-15 from: "While this can be estimated from spinning up a model that incorporates more reactive nitrogen species than just $NH_3$, we do not feel confident estimating total N input from modeled $NH_3$ dry deposition alone. At sites where total N input is known (e.g. BM, from Hurkuck et al. (2014), or from CTM results), the MNS and WK parameterizations predict such different $\Gamma_s$ estimates that one model predicts net emission from the stomata and one model predicts a net uptake over the course of the measurement campaign. A detailed investigation on the reasons for this mismatch is, however, beyond the scope of this paper."

to: "While this issue can be overcome by iteratively solving a model with more reactive nitrogen species, so that N input is both a parameter, and a result of the simulation, we here used a model that only predicts $NH_3$ dry deposition, which we do not consider to provide sufficient information to estimate total N input to our sites. At sites where total N input is known (e.g. BM, from Hurkuck et al. (2014), or from CTM results for other sites), the MNS and WK parameterizations both predict very different $\Gamma_s$ estimates. The reasons for this mismatch have, to our knowledge, not been investigated to date. We therefore decided to not model the stomatal pathway explicitly and rely on nighttime fluxes only."

Page 10 line 24-25: What do you mean by "very well"? Do you mean that the assumption of ground layer resistance = infinite is not realistic? And what about weak ground resistance and infinite stomatal resistance? The authors should give some more explanations and overall extract the main positive idea of such sensitivity tests explained in this paragraph. The reader is a bit frustrated not to know if good ideas have to be extracted from that.

$R_g = \infty$ is more of a technical solution to transform the model into a one-layer model in cases where ground-layer fluxes could not clearly be differentiated from other pathways when parameterizing the model on micrometeorological measurements, not necessarily based on whether or not it is realistic. In principle, it would of course be more realistic to model the ground-layer pathway for all land-use classes, even if weak ground-layer emissions are recaptured by the canopy (modeled via $\chi_c$) but according to Massad et al. (2010) not enough reliable data on ground-layer fluxes and emission potentials were available during the development of the parameterization. Note that one also has to be careful when mixing e.g. $R_w$ parameterizations based on micrometeorological measurements via inversion of a one-layer model together with measurements of $[NH_4^+]/[H^+]$ in the soil solution as an estimate for $\Gamma_g$, as the former will already include a contribution of the ground-layer when emission fluxes are large enough to not be completely recaptured by the canopy. Please also refer to our answer to P3L13.

**Changes to the manuscript:** P10L20-21: Replace "(…) the ground layer resistance was set to infinity (Massad et al., 2010) and the model reduces to a one-layer model." with "the ground layer resistance was set to infinity in order to transform the model structure to that of a one-layer model (Massad et al., 2010).

Page 10 line 28: The reader cannot understand the ideas mentioned in this 3.2 paragraph if the authors do not explain in what purpose they use moving averages of NH3 concentrations. What is the goal of this exercise?

We respectfully refer the reader to section 2.5 (starting on P7L22) of the manuscript, which we will slightly expand in the revised version of the manuscript. However, we would also like to use this discussion forum as a platform to elaborate on them in some more detail:

Most non-stomatal resistance parameterizations found in the literature are steady-state approximations of processes that we know to be dynamic by nature (e.g. Wentworth et al., 2016; Jones et al., 2007a, 2007b; and many others), i.e. they are solved for every moment in time individually, although we are aware that we should keep the history of the site in mind, especially with respect to previous nitrogen deposition and the wetness of the canopy. In the late nineties, Sutton et al. (1998) and Flechard et al. (1999) developed bidirectional cuticular desorption models that model the non-stomatal pathway as a charged capacitor, and they have been successful at modeling e.g. emission events after dewfall at night and subsequent drying of the canopy in the early morning hours. However, it turned out to be difficult to parameterize these models with measurements, e.g. surface pH, or concentration measurements of a number of different atmospheric constituents were needed. Additionally, as these models were naturally also dynamic in a numerical sense, i.e. dependent on the previous state of the

system (sometimes with very small time-steps needed for the numerical solution), they had the disadvantage of being computationally expensive, which limited their applicability in spatially explicit transport models. Wichink Kruit et al. (2010) presented an important step towards a balance between mechanistically realistic and computationally efficient models. They tried to simplify the bidirectional parameterization for external leaf surfaces by developing an external leaf surface compensation point model that was dependent on atmospheric ammonia concentrations, thus being capable of modeling saturation effects. While this approach was technically not really bidirectional, due to the fact that the best fit to the data was achieved with an expression which always yields an external leaf surface compensation point that is smaller than the ambient concentration, it led to an improvement in the way that we could now get good estimates for long-term net $NH_3$ deposition "for the right reasons", i.e. because sometimes there is a significant non-zero external leaf surface emission potential. However, this model only incorporated information about the current state of the system, not about the magnitude of previous deposition events or previous ambient $NH_3$ levels. In our manuscript, we tried to find a compromise between these two approaches by fundamentally following a similar approach to Wichink Kruit et al. (2010), but making it dependent on the past. We decided to use $NH_3$ concentration instead of (modeled) fluxes, as it is an easily accessible variable that is directly being measured instead of modeled and therefore available before any model calculations, and a direct driver of $NH_3$ saturation at humid surfaces. Reviewer #1 correctly pointed out that a logical next step would be to incorporate the "wetness history" of the site into such analyses, e.g. the average relative humidity of the previous day(s), or the days since the last rainfall. We here only presented the very first step towards a conceptually dynamic, but structurally static model of external leaf surface exchange, and while our results are not directly useful for modeling purposes, we believe they are a promising first step for the future treatment of the bidirectional non-stomatal pathway.

**Changes to the manuscript:** P7L25: Before "We here…" Add: "While it is capable of modeling saturation effects with an ambient ammonia concentration dependent non-stomatal compensation point, it only relies on $\chi_a$ at the current calculation step. A compromise between the truly dynamic models of Sutton et al. (1998) and Flechard et al. (1999) and the steady-state simplification of Wichink Kruit et al. (2010) would respect the site's history of reactive nitrogen inputs without falling back to a numerically dynamic model and, consequently, the same difficulties that limit the application of existing dynamic approaches in large-scale models, i.e. it would need to use a proxy for previous nitrogen deposition without relying on the model's flux predictions at an earlier calculation time. "

Page 11 line 2: why this case is not shown? It would have been interesting to see the results?

We opted for a linear horizontal axis in Figure 6 as the majority of the concentration data are in the sub-20 µg m$^{-3}$ region and the fact that there is a functional relationship is still obvious. Additionally, a linear least-squares fit to a log-transformed variable implicitly assumes a multiplicative error-model, the validity of which is unclear in this particular case. Mentioning it in P11L2 was primarily meant to give the reader an idea of the relationship, but there are probably more appropriate statistical models to show this.

The linear fit to the log-log-transformed data is shown in Figure 1 of this response.

[Figure]

*Figure 1: Log-log-relationship between moving averages of air ammonia concentrations with different moving windows and non-stomatal emission potentials derived from flux measurements at five sites.*

**Changes to the manuscript:** None.

Page 11 line 9 :The authors give indications of potential improvements and conclude by writing that no improvement is deduced. What is then the purpose of giving these results if they do not lead to improvement? It should be far better to highlight the advantages instead of giving the disadvantages.

This is probably more of a philosophical issue. The reviewer is correct in stating that the results of this particular attempt are not immediately useful. However, we believe that it is important to avoid publication bias by not exclusively showing polished up positive results, but also by publishing what would be considered 'negative' or 'non-constructive results'. Parameterizing $\Gamma_w$ on some proxy for site history is a logical step in reducing degrees of freedom of more complex, mechanistic models while still being conceptually (albeit not structurally/numerically) dynamic (see answer above). We tested one variant of doing so and arrived at the conclusion that this particular variant is probably not the final answer. By publishing these results regardless, we encourage looking at different approaches and avoid that other researchers unnecessarily try the same, only to arrive themselves at a 'negative', i.e. non-productive finding that they likely won't publish either. We feel that the manuscript at hand can stand on its own feet without this analysis, and if the reviewer and the editor agree that this section should be omitted from the manuscript, we are happy to do so, but we believe it adds some valuable information and gives the reader ideas on how (or how not) to improve $\Gamma_w$ parameterizations in the future.

**Changes to the manuscript:** None.

Page 11 line 20: same remark as above. The way this paragraph is written does not give a positive issue. The authors should turn it differently to highlight the positive points. This part should follow figure 3.

We have re-phrased the partial conclusion of this paragraph to appear more positive. However, we do not agree that this part should follow Fig. 3, as the motivation to reduce the parameter values of $R_{w,min}$ and $\beta$ follows from the left-hand panel of Fig. 5 (see answer below).

**Changes to the manuscript:** P11L17-20: Replace "While there does not appear to be a comprehensive, generic solution, we assume that there is potential for a significant overall improvement by optimizing these two parameters based on independent data from all four ecosystem types (grassland, arable, forest and semi-natural ecosystems) used in this parameterization." with "This exercise highlights the potential for a significant overall improvement in $NH_3$ flux predictions by optimizing these two parameters based on independent data from all four ecosystem types (grassland, arable, forest and semi-natural ecosystems) used in the MNS parameterization."

Page 12 line 1: What do you mean by "the impact of this study's main findings are negligible"?

We believe the reviewer might have misread this line (the manuscript says "on" instead of "of"), but we will rephrase this sentence for clarity.

**Changes to the manuscript:** Replace "(…) the impact on this study's main findings are negligible." with "(…) they do not lead to significant differences in the main findings of this study."

Page 12 line 12: Again what is the advantage of doing this if no solution is going out?

See answer to P11L9 given above.

**Changes to the manuscript:** None.

Page 12 line 13: the title is not appropriated. Should be "conclusions".

Agreed.

**Changes to the manuscript:** P12L12: Replace "Conclusions and recommendations for further research" with "Conclusions".

Page 12 line 17: "pollution climate" is not an understandable term. Conclusion needs to be more striking.

Regarding the term "pollution climate", see our reply to your comment on P5L24. We have re-formulated the second sentence to be more to the point.

**Changes to the manuscript:** P12L17-21: Replace "We tested the potential for an easily accessible improvement of predicted $R_w$ and consequently predicted $NH_3$ exchange fluxes by using smaller values for the temperature response and minimum $R_w$ parameters and propose to further investigate this route using data from all four ecosystem types represented in the MNS $R_w$ parameterization" with "Adjusting the temperature response and minimum $R_w$ parameters in the MNS model towards smaller values resulted in a better match between modeled and measured $NH_3$ fluxes at most, but not all sites. We suggest to further investigate the potential of re-calibrating these parameters to flux data from all four ecosystem types represented in the MNS $R_w$ parameterization. Compared to measured values found in the literature (e.g. Massad et al., 2010, Tab. 1), especially the minimum predicted $R_w$ at sites with low atmospheric acid-to-ammonia ratios appear too high."

Page 12 line 27: "We strongly encourage" is not appropriate. Please reformulate.

Agreed, this is probably too subjective.

**Changes to the manuscript:** P12L27: Replace "strongly encourage" with "suggest".

Changes in the structure are needed. Figure 7 should follow figure 3, figure 8 should follow figure 4. Please adapt the text in function of these figure changes.

We see where the reviewer is coming from, as these figures appear to be very similar visually. However, we respectfully disagree with this suggestion for the following reasons:

Figure 3 and Figure 7 are only similar in the form of the visualization. Figure 3 is a comparison between the two models' predicted fluxes in their original parameterization and marks the first step of our analysis. Figure 7, on the other hand, is the result of changing two parameters in a way that was suggested by the results. It answers a "what if" question that would not have been asked before seeing the left panel of Figure 5; it is not part of the "core" analysis of our manuscript, but more of an outlook, or a suggestion for what parameters to look at in the future. It would therefore not be logical to show it earlier.

Figure 8 is a sensitivity study with the aim to show the influence of some of our decisions and to assess "researcher's bias" introduced by making these choices. This was actually an Appendix in early versions of the manuscript, but since the paper itself is fairly short, we decided to move it to the results section instead. We are happy to move it back to an Appendix section if needed, but we don't think it should be shown earlier, as it is not a fundamental part of the analysis, but rather an addition.

**Changes to the manuscript:** None.

Technical corrections

Page 6 line 10 and line 26, ibid and i.e. have to be in italics. Throughout the text latin expressions should be in italics.

"Common Latin phrases are not italicized (for example, et al., cf., e.g., a priori, in situ, bremsstrahlung, and eigenvalue)."

From: www.atmospheric-chemistry-and-physics.net/for_authors/manuscript_preparation.html

**Changes to the manuscript:** None.

**References (both replies)**

Caird, M. A., Richards, J. H., and Donovan, L. A.: Nighttime Stomatal Conductance and Transpiration in $C_3$ and $C_4$ Plants, Plant Physiology, 143, 4–10, doi:10.1104/pp.106.092940, 2007.

Fisher, J. B., Baldocchi, D. D., Misson, L., Dawson, T. E., and Goldstein, A. H.: What the towers don't see at night: notcturnal sap flow in trees and shrubs at two AmeriFlux sites in California. Tree Physiology, 27(4), 596–610, doi:10.1093/treephys/27.4.597, 2007.

Flechard, C. R., Fowler, D., Sutton, M. A., and Cape, J. N.: A dynamic chemical model of bi-directional ammonia exchange between semi-natural vegetation and the atmosphere, Quarterly Journal of the Royal Meteorological Society, 125(559), 2611–2641, doi:10.1002/qj.49712555914, 1999.

Flechard, C. R., Spirig, C., Neftel, A., and Ammann, C.: The annual ammonia budget of fertilised cut grassland – Part 2: Seasonal variations and compensation point modeling, Biogeosciences, 7(2), 537–556, doi:10.5194/bg-7-537-2010, 2010.

Jones, M. R., Leith, I. D., Fowler, D., Raven, J. A., Sutton, M. A., Nemitz, E., Cape, J. N., Sheppard, L. J., Smith, R. I., and Theobald, M. R.: Concentration-dependent NH3 deposition processes for mixed moorland semi-natural vegetation, Atmospheric Environment, 41(10), 2049–2060, doi:10.1016/j.atmosenv.2006.11.003, 2007a.

Jones, M. R., Leith, I. D., Raven, J. A., Fowler, D., Sutton, M. A., Nemitz, E., Cape, J. N., Sheppard, L. J., and Smith, R. I.: Concentration-dependent NH3 deposition processes for moorland plant species with and without stomata, Atmospheric Environment, 41(39), 8980–8994, doi:10.1016/j.atmosenv.2007.08.015, 2007b.

Massad, R.-S., Nemitz, E., and Sutton, M. A.: Review and parameterisation of bi-directional ammonia exchange between vegetation and the atmosphere, Atmospheric Chemistry and Physics, 10, 10359–10386, doi:10.5194/acp-10-10359-2010, 2010.

Milford, C., Hargreaves, K. J., Sutton, M. A., Loubet, B., and Cellier, P.: Fluxes of $NH_3$ and $CO_2$ over upland moorland in the vicinity of agricultural land, Journal of Geophysical Research: Atmospheres, 106(D20), doi:10.1029/2001JD900082, 24169–24181, 2001.

Móring, A., Vieno, M., Doherty, R. M., Laubach, J., Taghizadeh-Toosi, A., and Sutton, M. A.: A process-based model for ammonia emission from urine patches, GAG (Generation of Ammonia from Grazing): description and sensitivity analysis, Biogeosciences, 13, 1837–1861, doi:10.5194/bg-13-1837-2016, 2016.

Nemitz, E., Sutton, M. A., Schjoerring, J. K., Husted, S., and Wyers, G. P.: Resistance modelling of ammonia exchange over oilseed rape, Agricultural and Forest Meteorology, 105(4), 405–425, doi:10.1016/S018-1923(00)00206-9, 2000.

Shen, J., Chen, D., Bai, M., Sun, J., Coates, T., Lam, S. K., and Li, Y.: Ammonia deposition in the neighbourhood of an intensive cattle feedlot in Victoria, Australia, Scientific Reports, 6, 32793, doi:10.1038/srep32793, 2016.

Sutton, M. A., Burkhardt, J. K., Guerin, D., Nemitz, E., and Fowler, D.: Development of resistance models to describe measurements of bi-directional ammonia surface–atmosphere exchange, Atmospheric Environment, 32(3), 473–480, doi:10.1016/S1352-2310(97)00164-7, 1998.

van Zanten, M. C., Sauter, F. J., Wichink Kruit, R. J., van Jaarsveld, J. A., and van Pul, W. A. J., Description of the DEPAC module; Dry deposition modeling with DEPAC_GCN2010, National Institute for Public Health and the Environment (RIVM), Bilthoven, The Netherlands, 2010.

Wentworth, G. R., Murphy, J. G., Gregoire, P. K., Cheyne, C. A. L., Tevlin, A. G., and Hems, R.: Soil–atmosphere exchange of ammonia in a non-fertilized grassland: measured emission potentials and inferred fluxes, Biogeosciences, 11, 5675–5686, doi:10.5194/bg-11-5675-2014, 2014.

Wentworth, G. R., Murphy, J. G., Benedict, K. B., Bangs, E. J., and Collett Jr., J. L.: The role of dew as a night-time reservoir and morning source for atmospheric ammonia, Atmospheric Chemistry and Physics, 16, 7435–7449, doi:10.5194/acp-16-7435-2016, 2016.

Wichink Kruit, R. J., van Pul, W. A. J., Sauter, F. J., van den Broek, M., Nemitz, E., Sutton, M. A., Krol, M., and Holtslag, A. A. M.: Modeling the surface–atmosphere exchange of ammonia, Atmospheric Environment, 44(7), 945–957, doi:10.1016/j.atmosenv.2009.11.049, 2010.

Wichink Kruit, R. J., Schaap, M., Sauter, F. J., van Zanten, M. C., and van Pul, W. A. J.: Modeling the distribution of ammonia across Europe including bi-directional surface–atmosphere exchange, Biogeosciences, 9, 5261–5277, doi:10.5194/bg-9-5261-2012, 2012.

Zhang, L., Brook, J. R., and Vet, R.: A revised parameterization for gaseous dry deposition in air-quality models, Atmospheric Chemistry and Physics, 3(6), 2067–2082, doi:10.5194/acp-3-2067-2003, 2003.

Zhang, L., Wright, L. P., and Asman, W. A. H.: Bi-directional air-surface exchange of atmospheric ammonia: A review of measurements and a development of a big-leaf model for applications in regional-scale air-quality models, Journal of Geophysical Research: Atmospheres, 115(D20), D20310, doi:10.1029/2009JD013589, 2010.

Zöll, U., Brümmer, C., Schrader, F., Ammann, C., Ibrom, A., Flechard, C. R., Nelson, D. D., Zahniser, M., and Kutsch, W. L.: Surface–atmosphere exchange of ammonia over peatland using QCL-based eddy-covariance measurements and inferential modeling, Atmospheric Chemistry and Physics, 16, 11283–11299, doi:10.5194/acp-16-11283-2016, 2016.

---

## Author Comment (AC2) · 12 Oct 2016

**Reply to Anonymous Referee #1**

The paper compares two alternative models to simulate ammonia fluxes and their comparison with the measured fluxes in five peatland and grassland sites, focusing on the non-stomatal fluxes. The motivation for such more empirical deposition models is the inclusion of bi-directional ammonia fluxes into chemical transport models, which requires few and easily available parameters for the surface resistance. The improvement of such models is needed and the respective analysis here is valuable, specifically as it includes a suffiecient number of sites.

We thank the Reviewer for his/her helpful comments and criticism and for valuing our work. We have implemented most of the suggestions and think they led to a significant improvement of our manuscript. Please refer to the point-by-point response below for a detailed reply to your comments.

Focusing exclusively on nighttime fluxes with sufficiently turbulent condition is a good approach. It should, however, be discussed, if nocturnal transpiration could have confounded these observations.

The assumption of an "infinite" stomatal resistance at night is indeed a strong simplification that is not necessarily physiologically correct, and we should have acknowledged this in the original manuscript. The reason behind it is that it allows modelers to easily differentiate between the stomatal and the non-stomatal pathway, depending on which of the two is assumed to dominate the other one in magnitude of the fluxes. These assumptions allow a very simple inversion of the one-layer model framework to derive $R_w$ (and $R_s$) from micrometeorological measurements, without having to explicitly model the other pathway. Consequences of this simplification are that $R_w$ derived in such a way may indeed partially integrate stomatal fluxes as well, and therefore the physiological meaning of this variable may be confounded. We will add a discussion of this in the uncertainties sub-section of the results.

**Changes to the manuscript:** P10L16 (will be moved to "Sources of uncertainty" sub-section; cf. reply to Reviewer #2): Add paragraph: "Explicitly modeling the stomatal pathway with physiologically accurate stomatal conductance models may have the additional benefit of being able to assess bias in the estimation of non-stomatal resistances introduced by nighttime stomatal opening, naturally resulting in a lower contribution of the non-stomatal pathway to the total observed flux. However, note that a distinction between physiological accuracy and the purpose which the derived resistances are used for has to be made. While nighttime stomatal opening is a well-known phenomenon (e.g. Caird et al., 2007), it is rarely respected in modeling studies (e.g. Fisher et al., 2007). A physiologically accurate $R_w$ parameterization used in conjunction with a stomatal model that does not account for nighttime stomatal opening would result in biased fluxes. We here derived $R_w$ under the assumption that stomata are closed at night to ensure comparability with $R_w$ values predicted by the WK and MNS parameterization, respectively, and compatibility with most operational biosphere-atmosphere exchange schemes, but we acknowledge that the physiological meaning may be confounded by stomatal flux contributions at night."; Add Caird et al. (2007) and Fisher et al. (2007) to list of references.

The parameters included in both models are temperature and, importantly, relative humidity (RH), whereas different ways are chosen regarding the fate of depositing ammonia, either unidirectional (MNS model) or 'quasi-bidirectional' (WK model). Based on common patterns of the five test sites, systematic under- and overestimation of fluxes are then diagnosed and empirical improvements are suggested by the authors. Although ultimately I agree with the overall direction of the paper and the interpretation of the results (see few exceptions below), I find it difficult to follow. One of the reasons is the continuous introduction of a multitude of parameters which makes consequent reading sometimes time-consuming and frustrating. Even in case it does not agree with the usual policy of this journal, I therefore suggest a table detailing all used parameters with units and possibly also other abbreviations.

We agree with the reviewer that constantly introducing new variables can interrupt the reader's flow. A list of symbols, on the other hand, can lead readers inexperienced with the modeling community's

jargon to skip back and forth between pages when new variables appear. Neither variant is very elegant in terms of an uninterrupted reading experience, but we are convinced that we here chose the lesser of two (necessary) evils, and that the manuscript would not benefit from using a list of symbols instead. The majority of all constants and variables are defined 'in passing', i.e. we introduce their respective symbols and units in an unobtrusive manner alongside their first appearance in the text (e.g. P3L21: "$R_c$ is further split into a stomatal pathway with the stomatal resistance $R_s$ (s m$^{-1}$), and […] the non-stomatal resistance $R_w$ (s m$^{-1}$) […].". There are only two slightly larger blocks of variable introductions after an equation: P4L10-13 (4 lines) and P5L1-3 (3 lines). All further shortening of the manuscript would be due to removing variable definitions less than one line long and the omission of units in the text. On the other hand, a complete list of symbols would have more than 40 entries, and a shortened list of symbols (e.g. defining only $R_x$ instead of $R_a, R_b, R_c, R_s, R_w$) would not be enough to omit variable definitions throughout the manuscript.

Additionally, units may change depending on the circumstances a symbol is used. In the modeling community, (compensation point-) concentrations are often given in µg m$^{-3}$ when they appear in figures, whereas some equations work on with concentrations in mol L$^{-1}$ (a well-known example is the traditional formulation for the conversion from emission potentials to compensation points, as seen e.g. in Nemitz et al., 2000). Explicitly stating or repeating units close to the appearance of a symbol helps avoid confusion in these cases.

As a compromise and as an additional resource for readers unexperienced with the modeling community's jargon, we have added a list of symbols in the supplementary material.

**Changes to the manuscript:** P3L13: add "For a list of variables used throughout this article, the reader is referred to Tab. S1 in the supplement."; add Table S1 (List of symbols) to the supplementary material.

A second and somewhat related reason is the initial explanation of the two models which in some important places is not sufficiently detailed – some examples are given below. I wonder why the effect of RH is so little discussed in the paper - it has an exponential influence on Rw and thus is the most important independent parameter. It could e.g. be included in a analysis similar to Fig. 5.

We agree that $RH$ is the most important parameter. However, we also think it has been sufficiently discussed elsewhere, and a simple exponential decay function does not necessarily need a visualization that goes beyond the relationship shown in Figure 2a. We will highlight the importance of the exponential decay function with minor changes to the text in some parts in the manuscript.

**Changes to the manuscript:** P2L18-19: Replace "This characteristic behavior is often modeled using relative humidity response functions as a proxy for canopy wetness (e.g. Sutton and Fowler, 1993; Erisman et al., 1994)." with "This characteristic behavior is typically modeled with an exponential relative humidity response function as a proxy for canopy wetness, where a high relative humidity results in low non-stomatal resistances and vice-versa (e.g. Sutton and Fowler, 1993; Erisman et al, 1994).". P5L3: Add "The exponential decay parameter $a$ was calculated as an average of $a$ values per land-use class reported in the literature". P5L14: Add "exponential" between "simple" and "humidity".

I also wonder if the effect of backward-looking moving averages shouldn't be evaluated together with the RH history. Saturation effects (as mentioned in p.2, l. 20) could play a role at low RH.

Agreed, this is a good idea for further analyses. It is not trivial to find a good balance between a truly dynamic, but demanding (both numerically and in terms of required input-data) representation of the non-stomatal pathway, and a steady-state simplification that can be incorporated easily into existing schemes. We here tried to go the very first step that is more or less as simple as possible (while still respecting site-history in some way), and we found that this is not enough. We hope that this is a valuable piece of information for future analyses, which could (and should) indeed incorporate

additional proxies, such as *RH* or precipitation history, but we deem this beyond the scope of our paper, where the moving-average approach was merely an additional idea about "what could work", even if it ultimately turned out to not work very well.

**Changes to the manuscript:** None (but added some explanation as suggested by Reviewer #2).

P. 5, l. 24/25: This is difficult to follow. Can it be supported by a formula? What happens if RH decreases?

We agree that it is not immediately obvious and tedious to show formally from Eqns. (7) and (8) that $\chi_w$ can only become a fraction of $\chi_a$, and we hope a visualization of the solution space over plausible $T$ and $\chi_a$ ranges (Figure 1 of this response) helps. Similar visualizations for possible $\Gamma_w$ and $\chi_w$ values can be found in van Zanten et al. (2010, Appendix F, Figs. 17 and 18), which we will refer to in the revised manuscript. A decrease in *RH* has no direct effect on modeled $\chi_w$, only on $R_{w,eff.}$ due to the exponential decrease in the 'clean-air' $R_w$ parameterization (Eq. (6)).

**Changes to the manuscript:** P5L25 add reference "(cf. van Zanten et al., 2010, Appendix F)." at the end of the sentence.

[Figure]

*Figure 1: Fractions of the solutions for Eq. (7) of the original manuscript divided by $\chi_a$ over a range of plausible $\chi_a$ and T values.*

P. 8, l. 19-23: This is indeed intriguing, but on the other hand I cannot really believe that MNS works so well in the prediction of fluxes at VK, when looking at the cumulative fluxes in Fig. 3. Even during the flat part, there is an underestimation of 0.3 kg ha-1. The shape of the cumulative fluxes at BM is considerably different from VK, while the shapes of ΔGw differences in Fig. 4 are very similar between BM and VK. Please check if the statement is really correct.

We did not state that MNS predicts the fluxes at VK well. "(…) relatively good predictive capabilities of MNS at BM and WK at VK (…)" (P8L21-22). We acknowledge that $\Delta G_w$ plots for the MNS model may indeed appear somewhat similar for BM and VK at first glance, but note that at VK we see a larger number of strong underestimations of $\Delta G_w$ compared to BM. The ratio of negative to positive values of $\Delta G_w$ for the MNS model is 1.2 at BM versus 2.8 at VK.

**Changes to the manuscript:** None.

P. 11, l. 19: which parameterization?

The MNS parameterization. Thanks for pointing out that this is unclear.

**Changes to the manuscript**: P11L19-20: Replace "(…) used in this parameterization." with "(…) used in the MNS parameterization.".

Figure 2: Why is Rw lowest at T=0°C?

Cf. P5L7-9: "Contrary to the original formulation of Flechard et al. (2010), Massad et al. (2010) do not use absolute values of $|T|$ (°C), but we chose to do so under the assumption that generally $R_w$ increases in freezing conditions (e.g. Erisman and Wyers, 1993)."

We here chose to follow the original formulation of this temperature correction by Flechard et al. (2010), as it seemed physically more plausible to us that ammonia deposition to liquid water is larger than to ice.

**Changes to the manuscript:** None (already explained in P5L7-9).

Figure 4: Upper row: There seems to be a mismatch between the number of binned values used for MNS and WK comparison, at least for VK. What is the reason?

The number of binned values per bin is of course different, as it defines the shape of the histograms. The bin-width is equal (100 s m$^{-1}$) for all figures shown in the upper row. Note that we cut-off everything below or above a -1000 or 1000 s m$^{-1}$ difference, respectively, for visual clarity and comparability of the subplots. So indeed the integral over the bars drawn in the figures does not necessarily reflect the total number of data points, if that is the question. The absolute differences can span multiple orders of magnitude, which we would not be able to visualize in a meaningful way. A logarithmic visualization would put more emphasis on the very large differences than necessary (e.g. there is not much difference between a $10^4$ and a $10^5$ s m$^{-1}$ difference between modeled and measured resistances other than the fact that it is an extremely large mismatch – either modeled or measured fluxes will be close to zero in both of these cases).

**Changes to the manuscript:** None.

Minor issues P.6, l.11: 'approach', better: 'reach'? (also p. 9, l. 2) P. 6, l. 16: 'compensation point Xw decreases' P. 6, l. 23: why 'moderately'? I would suggest to omit this word P. 7, l. 18: event P. 10, l. 1: omit 'and'

We agree with all corrections, thank you for pointing them out.

**Changes to the manuscript:** P6L11, P9L2: Replace "approach" with "reach". P6L16: Replace "compensation $\chi_w$ point" with "compensation point $\chi_w$". P6L22-23: Remove "moderately" in both hypotheses. P7L18: Replace "events" with "event". P10L1: Remove second "and" in the line.

**References (both replies)**

Caird, M. A., Richards, J. H., and Donovan, L. A.: Nighttime Stomatal Conductance and Transpiration in $C_3$ and $C_4$ Plants, Plant Physiology, 143, 4–10, doi:10.1104/pp.106.092940, 2007.

Fisher, J. B., Baldocchi, D. D., Misson, L., Dawson, T. E., and Goldstein, A. H.: What the towers don't see at night: notcturnal sap flow in trees and shrubs at two AmeriFlux sites in California. Tree Physiology, 27(4), 596–610, doi:10.1093/treephys/27.4.597, 2007.

Flechard, C. R., Fowler, D., Sutton, M. A., and Cape, J. N.: A dynamic chemical model of bi-directional ammonia exchange between semi-natural vegetation and the atmosphere, Quarterly Journal of the Royal Meteorological Society, 125(559), 2611–2641, doi:10.1002/qj.49712555914, 1999.

Flechard, C. R., Spirig, C., Neftel, A., and Ammann, C.: The annual ammonia budget of fertilised cut grassland – Part 2: Seasonal variations and compensation point modeling, Biogeosciences, 7(2), 537–556, doi:10.5194/bg-7-537-2010, 2010.

Jones, M. R., Leith, I. D., Fowler, D., Raven, J. A., Sutton, M. A., Nemitz, E., Cape, J. N., Sheppard, L. J., Smith, R. I., and Theobald, M. R.: Concentration-dependent NH3 deposition processes for mixed moorland semi-natural vegetation, Atmospheric Environment, 41(10), 2049–2060, doi:10.1016/j.atmosenv.2006.11.003, 2007a.

Jones, M. R., Leith, I. D., Raven, J. A., Fowler, D., Sutton, M. A., Nemitz, E., Cape, J. N., Sheppard, L. J., and Smith, R. I.: Concentration-dependent NH3 deposition processes for moorland plant species with and without stomata, Atmospheric Environment, 41(39), 8980–8994, doi:10.1016/j.atmosenv.2007.08.015, 2007b.

Massad, R.-S., Nemitz, E., and Sutton, M. A.: Review and parameterisation of bi-directional ammonia exchange between vegetation and the atmosphere, Atmospheric Chemistry and Physics, 10, 10359–10386, doi:10.5194/acp-10-10359-2010, 2010.

Milford, C., Hargreaves, K. J., Sutton, M. A., Loubet, B., and Cellier, P.: Fluxes of $NH_3$ and $CO_2$ over upland moorland in the vicinity of agricultural land, Journal of Geophysical Research: Atmospheres, 106(D20), doi:10.1029/2001JD900082, 24169–24181, 2001.

Móring, A., Vieno, M., Doherty, R. M., Laubach, J., Taghizadeh-Toosi, A., and Sutton, M. A.: A process-based model for ammonia emission from urine patches, GAG (Generation of Ammonia from Grazing): description and sensitivity analysis, Biogeosciences, 13, 1837–1861, doi:10.5194/bg-13-1837-2016, 2016.

Nemitz, E., Sutton, M. A., Schjoerring, J. K., Husted, S., and Wyers, G. P.: Resistance modelling of ammonia exchange over oilseed rape, Agricultural and Forest Meteorology, 105(4), 405–425, doi:10.1016/S018-1923(00)00206-9, 2000.

Shen, J., Chen, D., Bai, M., Sun, J., Coates, T., Lam, S. K., and Li, Y.: Ammonia deposition in the neighbourhood of an intensive cattle feedlot in Victoria, Australia, Scientific Reports, 6, 32793, doi:10.1038/srep32793, 2016.

Sutton, M. A., Burkhardt, J. K., Guerin, D., Nemitz, E., and Fowler, D.: Development of resistance models to describe measurements of bi-directional ammonia surface–atmosphere exchange, Atmospheric Environment, 32(3), 473–480, doi:10.1016/S1352-2310(97)00164-7, 1998.

van Zanten, M. C., Sauter, F. J., Wichink Kruit, R. J., van Jaarsveld, J. A., and van Pul, W. A. J., Description of the DEPAC module; Dry deposition modeling with DEPAC_GCN2010, National Institute for Public Health and the Environment (RIVM), Bilthoven, The Netherlands, 2010.

Wentworth, G. R., Murphy, J. G., Gregoire, P. K., Cheyne, C. A. L., Tevlin, A. G., and Hems, R.: Soil–atmosphere exchange of ammonia in a non-fertilized grassland: measured emission potentials and inferred fluxes, Biogeosciences, 11, 5675–5686, doi:10.5194/bg-11-5675-2014, 2014.

Wentworth, G. R., Murphy, J. G., Benedict, K. B., Bangs, E. J., and Collett Jr., J. L.: The role of dew as a night-time reservoir and morning source for atmospheric ammonia, Atmospheric Chemistry and Physics, 16, 7435–7449, doi:10.5194/acp-16-7435-2016, 2016.

Wichink Kruit, R. J., van Pul, W. A. J., Sauter, F. J., van den Broek, M., Nemitz, E., Sutton, M. A., Krol, M., and Holtslag, A. A. M.: Modeling the surface–atmosphere exchange of ammonia, Atmospheric Environment, 44(7), 945–957, doi:10.1016/j.atmosenv.2009.11.049, 2010.

Wichink Kruit, R. J., Schaap, M., Sauter, F. J., van Zanten, M. C., and van Pul, W. A. J.: Modeling the distribution of ammonia across Europe including bi-directional surface–atmosphere exchange, Biogeosciences, 9, 5261–5277, doi:10.5194/bg-9-5261-2012, 2012.

Zhang, L., Brook, J. R., and Vet, R.: A revised parameterization for gaseous dry deposition in air-quality models, Atmospheric Chemistry and Physics, 3(6), 2067–2082, doi:10.5194/acp-3-2067-2003, 2003.

Zhang, L., Wright, L. P., and Asman, W. A. H.: Bi-directional air-surface exchange of atmospheric ammonia: A review of measurements and a development of a big-leaf model for applications in regional-scale air-quality models, Journal of Geophysical Research: Atmospheres, 115(D20), D20310, doi:10.1029/2009JD013589, 2010.

Zöll, U., Brümmer, C., Schrader, F., Ammann, C., Ibrom, A., Flechard, C. R., Nelson, D. D., Zahniser, M., and Kutsch, W. L.: Surface–atmosphere exchange of ammonia over peatland using QCL-based eddy-covariance measurements and inferential modeling, Atmospheric Chemistry and Physics, 16, 11283–11299, doi:10.5194/acp-16-11283-2016, 2016.